# Imaging cytoplasmic lipid droplets in vivo with fluorescent perilipin 2 and perilipin 3 knock-in zebrafish

Meredith H Wilson[1], Stephen C Ekker[2], Steven A Farber[1,3]*

[1]Carnegie Institution for Science Department of Embryology, Baltimore, United States; [2]Department of Biochemistry and Molecular Biology, Mayo Clinic, Rochester, United States; [3]Johns Hopkins University Department of Biology, Baltimore, United States

**Abstract** Cytoplasmic lipid droplets are highly dynamic storage organelles that are critical for cellular lipid homeostasis. While the molecular details of lipid droplet dynamics are a very active area of investigation, this work has been primarily performed in cultured cells. Taking advantage of the powerful transgenic and in vivo imaging opportunities available in zebrafish, we built a suite of tools to study lipid droplets in real time from the subcellular to the whole organism level. Fluorescently tagging the lipid droplet-associated proteins, perilipin 2 and perilipin 3, in the endogenous loci permits visualization of lipid droplets in the intestine, liver, and adipose tissue. Using these tools, we found that perilipin 3 is rapidly loaded on intestinal lipid droplets following a high-fat meal and later replaced by perilipin 2. These powerful new tools will facilitate studies on the role of lipid droplets in different tissues, under different genetic and physiological manipulations, and in a variety of human disease models.

*For correspondence: farber@carnegiescience.edu

## Introduction

Cytoplasmic lipid droplets are cellular organelles composed of a core of neutral lipids surrounded by a monolayer of phospholipids and coated with a variety of proteins. While initially believed to be passive storage depots for lipids, it is now appreciated that lipid droplets are dynamic organelles with roles in cellular lipid homeostasis, protection from lipotoxicity and ER stress, viral and parasitic infection, and host defense (*Farese and Walther, 2009*; *Cloherty et al., 2020*; *Olzmann and Carvalho, 2019*; *Roberts and Olzmann, 2020*; *Bosch et al., 2020*).

Lipid droplets are typically coated by one or more perilipins (PLINs), an evolutionarily related protein family defined by two conserved protein motifs, the N-terminal ~100 amino acid hydrophobic PAT domain followed by a repeating 11-mer helical motif of varying length (*Kimmel and Sztalryd, 2016*; *Lu et al., 2001*; *Londos et al., 1999*; *Bussell and Eliezer, 2003*; *Miura et al., 2002*). PLINs are recruited to the lipid droplet surface directly from the cytosol, mediated at least in part by the 11-mer repeat regions which fold into amphipathic helices (*Rowe et al., 2016*; *Bulankina et al., 2009*; *Ajjaji et al., 2019*; *McManaman et al., 2003*; *Garcia et al., 2003*; *Giménez-Andrés et al., 2021*). There is also evidence that 4-helix bundles located in the C-terminus regulate the affinity and stability of lipid droplet binding (*Ajjaji et al., 2019*; *Rowe et al., 2016*; *Hickenbottom et al., 2004*; *Titus et al., 2021*; *Chong et al., 2011*). PLINs act to regulate lipid storage by preventing or modulating access of lipid droplets to lipases and lipophagy machinery (*Sztalryd and Brasaemle, 2017*; *Granneman et al., 2007*; *Wang et al., 2011b*; *Wang et al., 2009*; *Patel et al., 2014*; *Granneman et al., 2009*; *Listenberger et al., 2007*; *Kaushik and Cuervo, 2015*).

PLINs have been found in species ranging from *Dictyostelium* to mammals, with more divergent variants in fungi and *Caenorhabditis* (*Gao et al., 2017*; *Miura et al., 2002*; *Wolins et al., 2003*; *Greenberg et al., 1991*; *Jiang and Serrero, 1992*; *Brasaemle et al., 1997*; *Chughtai et al., 2015*; *Lu et al., 2001*; *Sun et al., 2015*; *Wang and St Leger, 2007*; *Díaz and Pfeffer, 1998*; *Yamaguchi et al., 2006*). The human genome contains five *PLIN* genes, now designated *PLIN*1–5 (*Kimmel et al., 2010*). PLIN1 is predominantly expressed in white and brown adipoctyes (*Greenberg et al., 1991*), whereas PLIN2 and PLIN3 are expressed ubiquitously (*Brasaemle et al., 1997*; *Heid et al., 1998*; *Díaz and Pfeffer, 1998*; *Wolins et al., 2001*). PLIN4 is found in adipocytes, brain, heart, and skeletal muscle, and PLIN5 is located in fatty acid oxidizing tissues such as heart, brown adipose tissue, and skeletal muscle, as well as in the liver (*Dalen et al., 2007*; *Wolins et al., 2006*; *Yamaguchi et al., 2006*). The genomes of ray fin fish, including zebrafish, have orthologs of human PLIN1, PLIN2, and PLIN3, in addition to the unique PLIN variant, perilipin 6 (Plin6), which targets the surface of pigment-containing carotenoid droplets in skin xanthophores (*Granneman et al., 2017*; *Huang et al., 2020*).

While fluorescently tagged PLIN reporter proteins are used extensively in cell culture to visualize lipid droplets (e.g., *Miura et al., 2002*; *Targett-Adams et al., 2003*; *Kaushik and Cuervo, 2015*; *Granneman et al., 2017*; *Chung et al., 2019*; *Schulze et al., 2020*), in vivo lipid droplets have been historically studied in fixed tissues using immunohistochemistry (*Frank et al., 2015*; *Lee et al., 2009*), staining with lipid dyes (Oil Red O, Sudan Black, LipidTox), or by electron microscopy (*Chughtai et al., 2015*; *Zhang et al., 2010*; *Marza et al., 2005*). Lipid droplets can also be labeled in live organisms with fluorescent lipophilic dyes such as BODIPY (*Masedunskas et al., 2017*) and Nile red (*Minchin and Rawls, 2017a*), fed with fluorescently tagged fatty acids (BODIPY or TopFluor), which are synthesized into stored fluorescent triglycerides or cholesterol esters (*Carten et al., 2011*; *Ashrafi et al., 2003*; *Furlong et al., 1995*; *Quinlivan et al., 2017*), or imaged in the absence of any label using CARS or SRS microscopy (*Chughtai et al., 2015*; *Chien et al., 2012*; *Wang et al., 2011a*). However, expression of fluorescently tagged lipid droplet-associated proteins in vivo has been primarily limited to yeast (*Gao et al., 2017*), *Drosophila* (*Grönke et al., 2005*; *Bi et al., 2012*; *Beller et al., 2010*), and *Caenorhabditis elegans* (*Zhang et al., 2010*; *Chughtai et al., 2015*, *Xie et al., 2019*), although a transgenic zebrafish Plin*2-tdtomato* line was recently described (*Lumaquin et al., 2021*).

Here, we report the generation and validation of zebrafish PLIN reporter lines, including *Fus(EGFP-plin2)* and *Fus(plin3-RFP)*, in which we inserted fluorescent reporters in-frame with the coding sequence at the genomic loci. These reporter lines faithfully recapitulate the endogenous expression and transcriptional regulation of *plin2* and *plin3* in larval zebrafish, allowing for in vivo imaging of lipid droplet dynamics in live animals at the subcellular, tissue, organ, and whole larvae level. Using these lines, we describe the ordered recruitment of Plin3 and then Plin2 to lipid droplets in intestinal enterocytes following the consumption of a high-fat meal. We also reveal a delay in hepatic expression of Plin2 and Plin3 during development and identify a population of Plin2-positive lipid droplets adjacent to neuromasts in the posterior lateral line.

## Results

### *plin2* and *plin3* are expressed in the intestine of larval zebrafish

To determine the tissue localization of *plin2* and *plin3* mRNA expression in larval zebrafish, we performed whole-mount in situ hybridization. Our data indicate that *plin2* is not expressed in any tissues of unfed larvae at 6 days post fertilization (dpf) (*Figure 1A and B*). However, following a high-fat meal, *plin2* mRNA expression is strongly induced in the intestine, consistent with findings in mice (*Lee et al., 2009*) and with our previous RNAseq and qRT-PCR data (*Zeituni et al., 2016*). *Plin3* mRNA is present in the intestines of both unfed and fed larvae, and the signal in the intestine is stronger in larvae fed a high-fat meal (*Figure 1A and B*), again consistent with previous data in fish and mice (*Zeituni et al., 2016*; *Lee et al., 2009*). Surprisingly, neither *plin2* nor *plin3* mRNA expression was noted in the liver or in other tissues of unfed or high-fat fed zebrafish larvae at 6 dpf (*Figure 1A and B*) as was expected from studies in mouse and human tissues (*Brasaemle et al., 1997*; *Heid et al., 1998*; *Than et al., 1998*; *Wolins et al., 2006*).

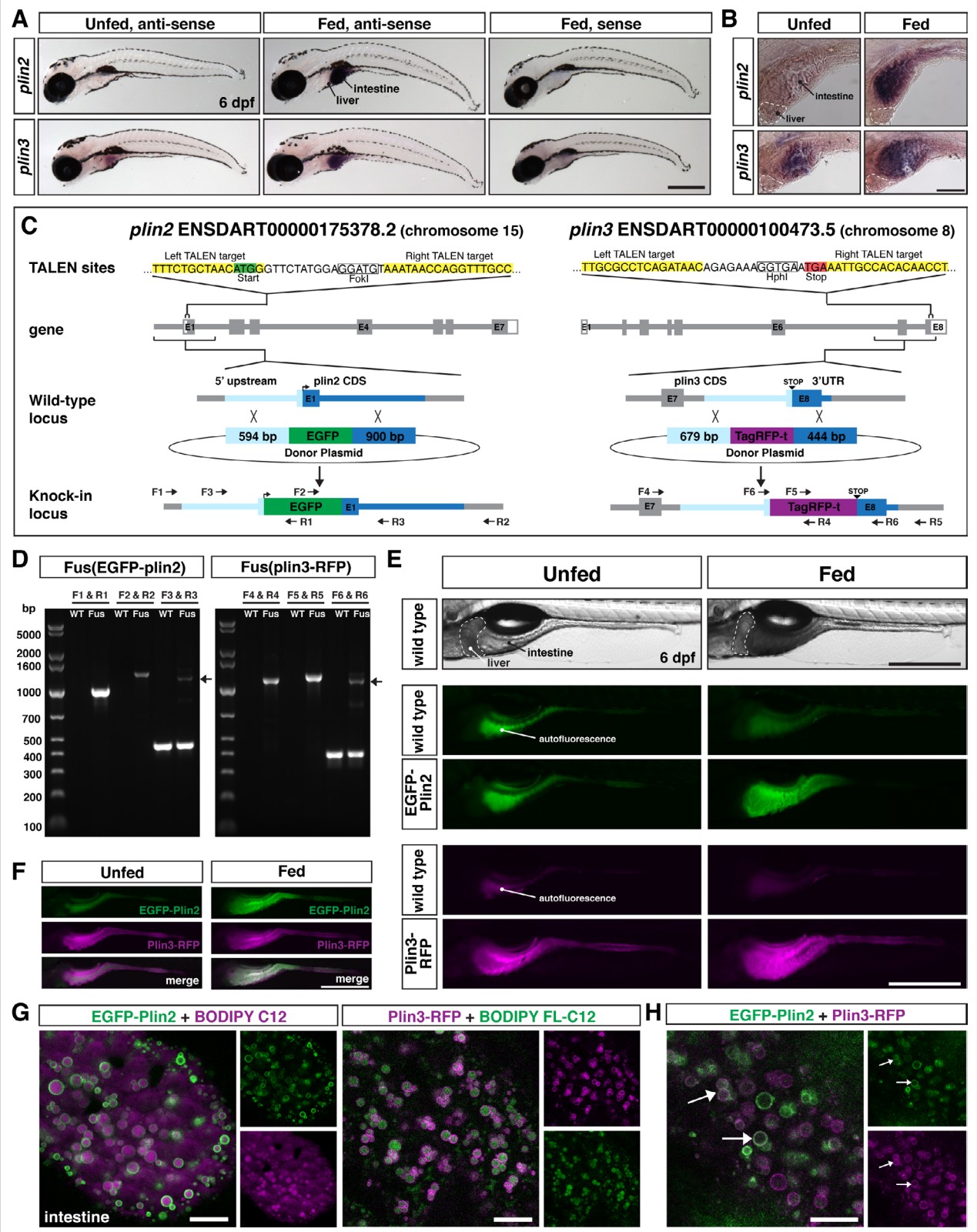

**Figure 1.** Generation of in-frame fluorescent reporters in the endogenous *plin2* and *plin3* loci. (**A,B**) Representative images of whole-mount in situ hybridization (ISH) with probes against zebrafish *plin2* (*ENSDARG00000042332*) and zgc: 77,486 (*plin3/4/5*) (*ENSDARG00000013711*) at 6 days post fertilization (dpf) either unfed or following feeding with a high-fat meal for 90 min. ISH was performed three times for each gene with n = 10 larvae per probe per experiment; scale = 500 μm (**A**), scale = 100 μm (**B**). *Plin2* is expressed in the intestine only following a high-fat meal whereas *plin3*

*Figure 1 continued on next page*

*Figure 1 continued*

is expressed in the intestine in unfed fish and has stronger expression following a high-fat meal. (**C**) Overview of the location and strategy used for TALEN-mediated genome editing. EGFP was fused in-frame at the N-terminus of *plin2*. TALEN targets in *plin2* are located in exon 1 of the *plin2-203 ENSDART00000175378.2* transcript and flank a FokI restriction site, loss of which was used to confirm cutting activity. A donor plasmid with the coding sequence for EGFP and *plin2* homology arms was co-injected with TALEN mRNA into one-cell stage embryos to be used as a template for homology directed repair. mTag-RFP-t was fused in-frame at the C-terminus of *plin3*. TALEN targets were located in exon 8 of the *plin3 ENSDART00000100473.5* transcript and flank the termination codon and an HphI restriction site, loss of which was used to confirm cutting activity. A donor plasmid with the coding sequence for mTagRFP-t and *plin3* homology arms was co-injected with TALEN mRNA into one-cell stage embryos to be used as a template for homology directed repair. (**D**) Following identification of fluorescent embryos in the F1 generation, RT-PCR and sequencing of genomic DNA using the primers noted on the knock-in loci depicted in (**C**) were used to confirm successful in-frame integration of the fluorescent tags. The size of the amplicons expected for correct integration were as follows: F1–R1 1033 bp, F2–R2 1340 bp, F3–R3 440 bp for wild-type (WT) and 1224 bp for *Fus(EGFP-plin2)* fusion, F4–R4 1218 bp, F5–R5 1274 bp, F6–R6 401 bp for WT and 1187 for *Fus(plin3-RFP)*. Arrows indicate the larger amplicon in heterozygous fish carrying the fusion alleles. (**E**) Imaging in live larvae (6 dpf) reveals expression of EGFP-Plin2 only in the intestine of larvae fed a high-fat meal (7 hr post-start of 2 hr meal) and Plin3-RFP is expressed in the intestine of both unfed and fed larvae (4.5 hr post-start of 2 hr meal, larvae are heterozygous for the fusion proteins; the lumen of the intestine has strong autofluorescence in WT and transgenic fish; see *Figure 1—figure supplement 2* for images of whole fish). Scale = 500 µm. (**F**) Examples of larvae expressing both EGFP-Plin2 and Plin3-RFP (7 hr post start of meal). Scale = 500 µm. For (**E and F**), images are representative of at least 15 fish from three independent clutches. (**G**) EGFP-Plin2 (green) and Plin3-RFP (magenta) label the lipid droplet surface in the intestine of 6 dpf larvae fed with a high-fat meal containing either BODIPY 558/568-C12 (magenta) or BODIPY FL-C12 (green) to label the stored lipids. Note the 558/568-C12 is not fully incorporated into stored lipid and is also found diffuse in the cytoplasm. Scale = 10 µm. (**H**) EGFP-Plin2 and Plin3-RFP can decorate the same lipid droplets in the intestine. Arrows denote examples of dual-labeled droplets, scale = 10 µm.

The online version of this article includes the following figure supplement(s) for figure 1:

**Figure supplement 1.** A deletion upstream of exon 1 in *plin2* impacts gene expression following a high-fat meal.

**Figure supplement 1—source data 1.** Source data associated with *Figure 1—figure supplement 1* panel D.

**Figure supplement 2.** Whole fish images corresponding to *Figure 1E*.

## Generation of knock-in lines to study lipid droplets in vivo

To study how Plin2 and Plin3 regulate lipid droplet dynamics in vivo, we generated fluorescent Plin2 and Plin3 zebrafish reporter lines (*Figure 1C*). We recognized that over-expression of Plin proteins can alter lipid droplet dynamics by altering the rate of lipolysis and lipophagy, which can result in altered levels of cytoplasmic lipid and lipoprotein secretion (*Bell et al., 2008*; *Listenberger et al., 2007*; *Fukushima et al., 2005*; *Bosma et al., 2012*; *McIntosh et al., 2012*; *Tsai et al., 2017*; *Magnusson et al., 2006*). Therefore, we chose to tag the endogenous proteins by engineering knock-in alleles.

We designed our engineering strategy for *Fus(EGFP-plin2)* based on early data suggesting that it was necessary to tag PLIN2 on the N-terminus (*Targett-Adams et al., 2003*), although subsequent work has now shown that C-terminal tags are also functional (*Kaushik and Cuervo, 2015*; *Lumaquin et al., 2021*). Additionally, we had evidence from our generation of Tol2-based transgenic reporter lines that over-expression of both human EGFP-PLIN2 and PLIN3-EGFP resulted in labeling of lipid droplets (refer to *Figure 4—figure supplement 1* and Figure 8). We used TALENs to introduce a double-strand break near the start codon in exon 1 in *plin2* (*ENSDART00000175378.2*) and adjacent to the termination codon in the last exon of *plin3* (*ENSDART00000100473.5*). One-cell stage embryos were injected with TALEN mRNA and donor constructs including either *EGFP* for *plin2* or *tagRFP-t* for *plin3*, flanked by the noted homology arms to direct homology directed repair (*Figure 1C*). The left homology arm for *plin2* included the 54 bp variable sequence we discovered upstream of exon 1, which may contain a regulatory element for control of *plin2* expression (*Figure 1—figure supplement 1*).

From the injected F0 adult fish, we identified founders for each knock-in allele by out-crossing and screening progeny for fluorescence in the intestine at 6 dpf either following a meal (*Fus(EGFP-plin2)*) or prior to feeding (*Fus(plin3-RFP)*). Correct integration of the fluorescent constructs was confirmed by PCR on genomic DNA of individual larvae, followed by sequencing (*Figure 1D*). For *Fus(EGFP-plin2)*, a total of 40 adults were screened and three fish produced EGFP+ progeny. However, two of these clutches had abnormal tissue expression patterns and only displayed cytoplasmic EGFP fluorescence. One of these had correct integration across the left homology arm, but the right homology arm integration could not be verified; the second had no confirmed integration at the locus. Furthermore, while our founder fish produced low numbers of progeny with correct expression patterns and proper integration (8 embryos out of 853 screened), she also produced progeny with incorrect integration

(22/853), suggesting mosaicism in the germ cells. To identify a *Fus(plin3-RFP)* founder, 26 adults were screened before a fish produced progeny expressing RFP. These embryos had correct tissue expression of RFP in the intestine and correct integration of the transgene (53 RFP+ embryos out of 613 screened). We did note a 29 bp region missing in intron 7–8; however, this does not alter the coding region, and it is unclear whether this deletion arose during integration or whether it was a naturally occurring variant in the injected embryo.

Consistent with our whole-mount in situ hybridization data, we observe EGFP-Plin2 only in the intestine of knock-in larvae fed a high-fat meal, whereas Plin3-RFP is detected in the intestine of both unfed and fed larvae (*Figure 1E*). No fluorescence is noted in the liver or in other tissues at 6 dpf, regardless of feeding status (for images of whole fish, see *Figure 1—figure supplement 2*). Fish carrying the knock-in alleles can be in-crossed and resulting progeny express RFP-Plin3 in the intestine prior to feeding and both transgenes are expressed subsequent to consuming a high-fat meal (*Figure 1F*). Thus, these knock-in alleles faithfully recapitulate the endogenous tissue mRNA expression patterns of *plin2* and *plin3* as revealed by in situ hybridization of fixed larval zebrafish (*Figure 1A*).

## EGFP-Plin2 and Plin3-RFP decorate lipid droplets in intestinal enterocytes

Enterocytes, the polarized epithelial cells that line the intestine, are responsible for absorbing and packaging dietary fat, primarily as triacylglycerol, into chylomicrons for export to the tissues of the body or cytoplasmic lipid droplets for storage (*Demignot et al., 2014*). Therefore, to confirm that the fluorescently tagged Plin2 and Plin3 proteins in our knock-in lines decorate the surface of lipid droplets, we performed confocal imaging in the anterior intestine of live larvae following a high-fat meal. *Fus(EGFP-plin2)/+* larvae were fed liposomes containing the fluorescent fatty acid analog BODIPY 558/568-C12. This fatty acid analog can be incorporated into both phospholipid and triglycerides for storage in lipid droplets in larval zebrafish (*Quinlivan et al., 2017*). As shown in *Figure 1G*, EGFP-Plin2 decorates the surface of BODIPY 558/568-C12-positive lipid droplets in the intestine, which is visualized as rings of EGFP fluorescence in a single confocal z-plane. Similarly, Plin3-RFP localizes to the surface of intestinal lipid droplets labeled with the green fluorescent BODIPY FL-C12 fatty acid analog (*Figure 1G*). In larvae heterozygous for both *Fus(EGFP-plin2)* and *Fus(plin3-RFP)*, we found that lipid droplets can be labeled by both EGFP-Plin2 and Plin3-RFP proteins (*Figure 1H*). This data confirms that both knock-in lines serve as valid fluorescent lipid droplet reporters amenable for live imaging at the organelle level. While resolving cell membranes in the anterior intestine of larvae is difficult with brightfield or differential interference contrast microscopy due to the three-dimensional nature of the intestinal folds and small cell size, these PLIN reporter lines could be crossed to fish carrying transgenic markers of the cell membranes (*Alvers et al., 2014*) to better elucidate the localization of lipid droplets within individual enterocytes.

## Intestinal Plin protein dynamics following a lipid-rich meal

Lipid droplets in the intestine are very dynamic. No lipid droplets are found in enterocytes in fasted mice, but in the initial hours following a high-fat meal, lipid droplets grow in number and size, and then are nearly depleted 12 hr later (*Zhu et al., 2009*). Despite an understanding of the dynamics, the localization of PLIN2 and PLIN3 on lipid droplets in enterocytes in response to high-fat feeding remains poorly understood. Prior studies in mice have suggested that PLIN3 is located on lipid droplets following an acute high-fat meal, but not following chronic high-fat feeding (*Lee et al., 2009*; *D'Aquila et al., 2015*). In contrast, despite upregulation of PLIN2 protein in the mouse intestine following an acute feed, PLIN2 was only present on lipid droplets in enterocytes following chronic high-fat feeding (*Lee et al., 2009*; *D'Aquila et al., 2015*). However, these findings were based on single time-points and no studies have imaged PLINs in the live vertebrate intestine at the level possible with our transgenic zebrafish lines. Therefore, to provide a more comprehensive understanding of PLIN expression and localization in the intestine following consumption of dietary fat, we performed imaging studies at both the whole intestine and tissue/cell/organelle level using the fluorescent knock-in reporter lines.

To start, we obtained whole-mount images of the intestine following a meal and quantified the fluorescence intensity over time (*Figure 2A and B*). The Plin3-RFP fluorescence, which is present in unfed fish, maximally increases approximately twofold, occurring ~14–16 hr after the onset of the meal.

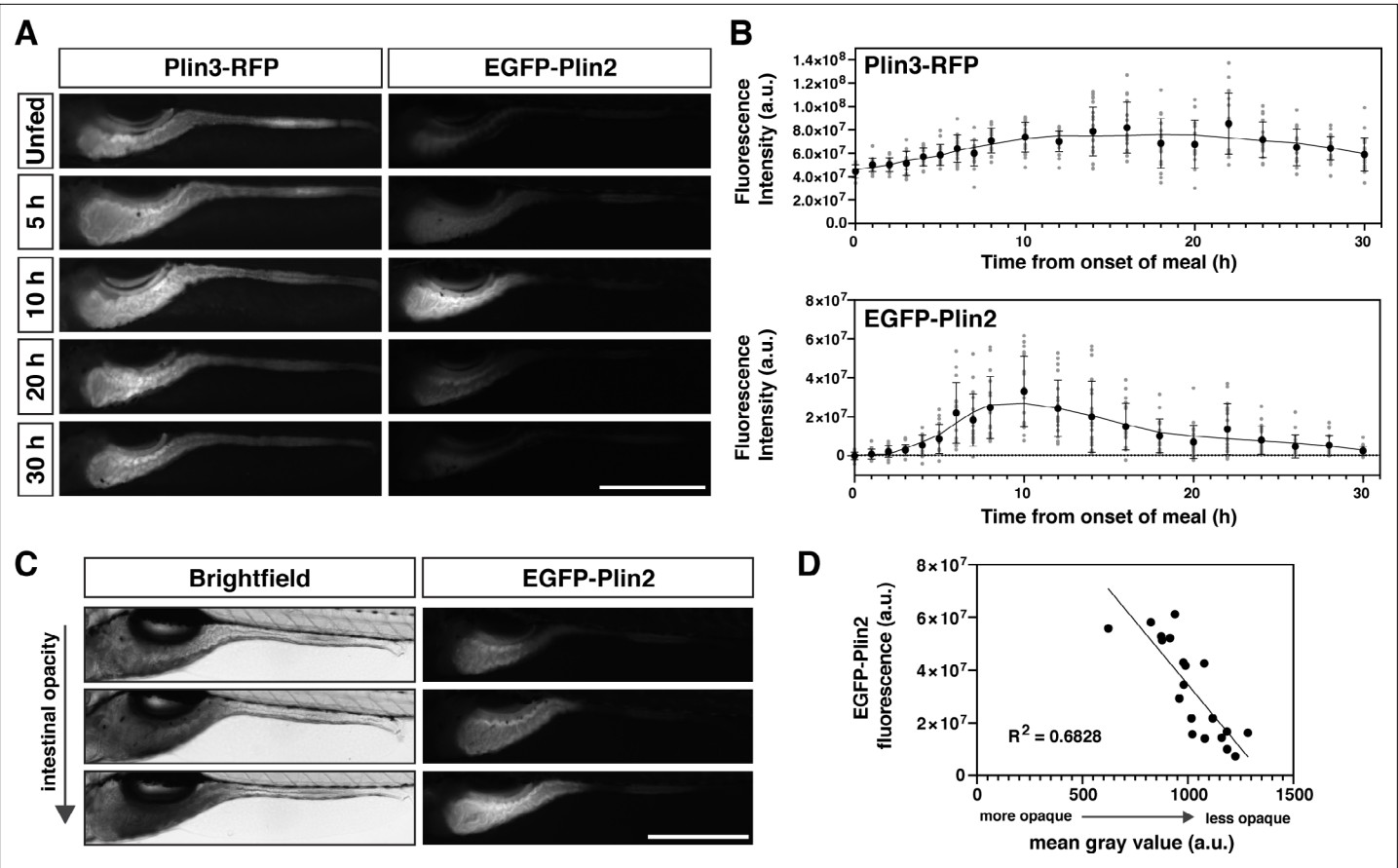

**Figure 2.** Whole-mount Plin protein expression in the intestine following a high-fat meal. (**A**) Representative images of Plin3-RFP and EGFP-Plin2 fluorescence in the intestine of 6 days post fertilization (dpf) larvae at noted time-points following the onset of a 90 min high-fat meal. Scale = 500 μm. (**B**) Fluorescence intensity was quantitated every hour for the first 8 hr following onset of the meal and subsequently every 2 hr for a total of 30 hr. Mean fluorescence intensity of wild-type siblings at each time-point was used to correct for gut autofluorescence. Black circles and errors bars indicate mean ± SD, individual data points are shown in gray, n = 14–23 total larvae per time-point from three independent clutches. Significant changes in fluorescence between unfed (0 hr) and subsequent time-points were calculated with Kruskal-Wallis with Dunn's multiple comparisons tests. For Plin3-RFP, fluorescence is significantly different from 0 hr at 6 and 8–28 hr, p < 0.05. For EGFP-Plin2, fluorescence is significantly different from 0 hr at 5–18, 22, and 24 hr, p < 0.05. Refer to *Figure 2—source data 1* file for p-values at each time-point. (**C**) Example brightfield images and corresponding EGFP-Plin2 fluorescence images from larvae imaged 10 hr after the onset of the meal. The degree of intestinal opacity reflects the amount of lipid consumed. Scale = 500 μm. (**D**) EGFP-Plin2 fluorescence intensity of individual larvae 10 hr after the onset of the meal was plotted as a function of the mean gray value of the intestine in the corresponding brightfield image. The amount of lipid consumed (intestine opacity) predicts much of the EGFP-Plin2 fluorescence (simple linear regression, y = –96314x + 130885876, $R^2$ = 0.6828, p < 0.0001).

The online version of this article includes the following figure supplement(s) for figure 2:

**Source data 1.** Source data associated with *Figure 2B and D*.

In contrast, the fluorescence intensity of EGFP-Plin2 increases ~20 million-fold, peaking at 8–12 hr. Furthermore, there is substantial variability in the fluorescence intensity of individual *Fus(EGFP-plin2)/+* fish at any given time-point (*Figure 2C*), which can be explained, in part, by differences in the quantity of lipid consumed (*Figure 2D*).

## Temporal dynamics of EGFP-Plin2 and Plin3-RFP in live intestinal enterocytes

PLIN3 is expressed throughout the cytoplasm in cells, including enterocytes, in the absence of lipid droplets and binds to nascent lipid droplets as they emerge from the endoplasmic reticulum (*Wolins et al., 2005*; *Bulankina et al., 2009*; *Skinner et al., 2009*; *Chung et al., 2019*; *Lee et al., 2009*). In contrast, PLIN2 is only stable when bound to lipid droplets and is quickly ubiquitinated and degraded in the absence of droplets (*Xu et al., 2005*; *Takahashi et al., 2016*; *Nguyen et al., 2019*; *Masuda*

*et al., 2006*). In keeping with these data, we find that in unfed larval zebrafish, in the absence of lipid droplets, Plin3-RFP is distributed throughout the cytoplasm of the intestinal enterocytes (*Figure 3A and B*, unfed). After consuming a high-fat meal, the Plin3-RFP pattern changes considerably: there is less cytoplasmic signal and bright puncta are present, which likely correspond to clusters of small lipid droplets (1.5 hr). Conversely, weak EGFP-Plin2 signal, almost exclusively on lipid droplets, only emerges ~2–3 hr after the start of a meal, which is consistent with the need to both transcribe (*Figure 1A* and *Zeituni et al., 2016*) and translate the protein after the start of the meal. The timing is also consistent with our measurements of EGFP fluorescence at the organ level (*Figure 2B*). As time continues, the EGFP-Plin2 signal on droplets increases strongly, whereas the Plin3-RFP fluorescence becomes predominantly cytoplasmic again by 6–7 hr.

To try to better understand the overlap of Plin3-RFP and EGFP-Plin2 on lipid droplets, we performed a fluorescence colocalization analysis in which we assessed the Manders' colocalization coefficients (tM1 and tM2) (*Figure 3C*). While an analysis of a subset of images suggests an increase in fluorescence colocalization over time following the meal, this analysis does not provide any information on the location of the colocalized pixels (cytoplasm vs. lipid droplet surface). Furthermore, because Plin3 is always present throughout the cytoplasm and translocates to lipid droplets, it will always colocalize with all EGFP-Plin2-positive pixels unless the cytoplasmic expression is thresholded to zero. The auto-threshold calculations in the algorithm will only partially correct for this issue, calling into question the usefulness of this metric for comparing proteins with these types of expression patterns. Therefore, we performed more comprehensive time-course experiments in larvae fed with 5 % egg yolk containing either the red or green fluorescent BODIPY C12 fatty acid analog, allowing us to visualize both the lipid within the enterocytes in addition to the fluorescent Plin proteins (*Figure 4A*). Since the fluorescent fatty acid accumulates in lipid droplets, it allows for more accurate segmentation of the droplets and subsequent quantitation of Plin localization (*Figure 4B*).

Within 45 min of meal onset, the green BODIPY FL-C12 is present within the intestinal lumen and throughout the cytoplasm of the enterocytes. It is also incorporated into lipids stored within nascent lipid droplets, which appear in puncta that are modestly brighter than the surrounding cytoplasm (*Figure 4A*, *Figure 4—video 1*, *Figure 4—video 2*, *Figure 4—video 3*). Over the next few hours, the lipid droplets grow in size and the contrast between the BODIPY signal in the lipid droplet vs. the cytoplasm is enhanced. In many fish, lipid droplets continue to be present in the enterocytes 30 hr following the onset of the meal, although their numbers are greatly reduced. While this same general trend occurs in fish fed with the red BODIPY C12 558/568 analog, the cytoplasmic signal remains high for much longer, resulting in much less contrast between the lipid droplets and surrounding cytoplasm over the first 8 hr (*Figure 4A*).

Consistent with our previous imaging (*Figure 3A and B*), the Plin3-RFP, which is already present in the cytoplasm in the absence of food, associates with the BODIPY-positive lipid droplet puncta as soon as 45 min (Plin3 on lipid droplets vs. cytoplasm, $p < 0.0001$; two-way ANOVA with Šídák multiple comparisons test), the first time-point we could reliably image after feeding. Plin3 continues to visibly coat lipid droplets for ~6 hr, although the mean fluorescence intensity on the droplets is declining over this time (correlation coefficient, $R = -0.3266$, *Figure 4C*). Despite the continued presence of lipid droplets, between 16 and 30 hr, there is no change in Plin3 fluorescence intensity associated with the droplets ($R = 0.0349$) and it is not significantly different from the Plin3 signal in the cytoplasm ($p > 0.99$; two-way ANOVA with Šídák multiple comparisons test). In contrast, the EGFP-Plin2 signal is first visibly detected in the enterocytes ~2–3 hr. While the EGFP-Plin2 fluorescence associated with lipid droplets increases substantially from 45 min to 8.5 h ($R = 0.7732$, *Figure 4D*), the first time-point at which it is statistically different from the cytoplasm is at 5.5 hr ($p = 0.0103$; two-way ANOVA with Šídák multiple comparisons test). Plin2 continues to decorate all of the droplets persisting through 30 hr after the onset of the meal, although the fluorescence intensity declines with time from 16 to 30 hr ($R = -0.6321$). Thus, the quantitative analysis of RFP and EGFP fluorescence associated with the lipid droplets vs. in the cytoplasm confirms the progression from Plin3 to Plin2 on the lipid droplet surface over time following a high-fat meal (*Figure 4B–D*). When the fluorescence intensity is normalized and the data is overlaid, the transition between the initial Plin3 coat to the longer-lasting Plin2 coat becomes even more evident (*Figure 4F*).

Based on this analysis, we wondered whether the timing of the transition to Plin2 simply reflects the time needed to both transcribe and translate Plin2 prior to its association with the droplets,

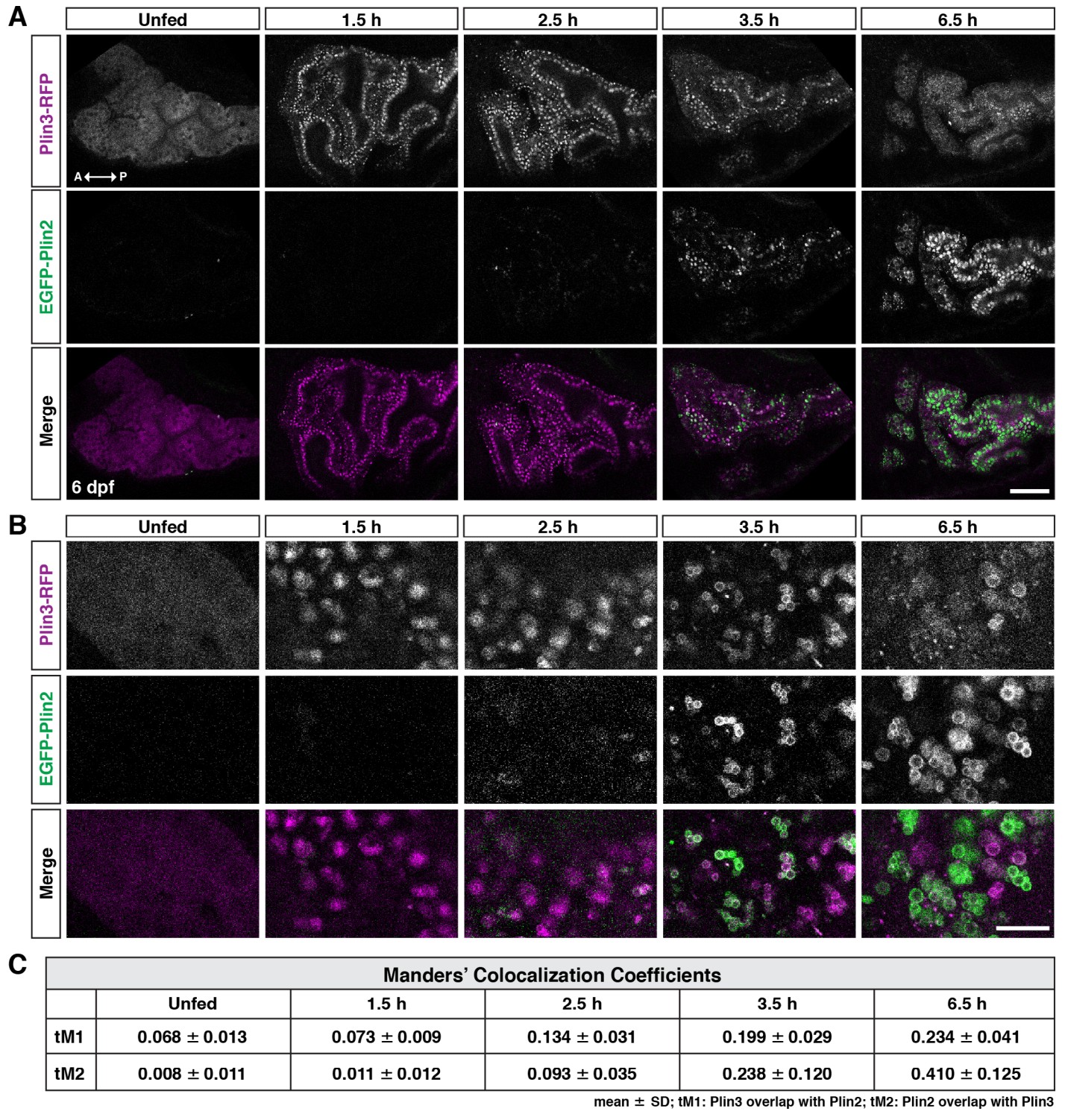

mean ± SD; tM1: Plin3 overlap with Plin2; tM2: Plin2 overlap with Plin3

**Figure 3.** Following a high-fat meal, Plin3-RFP and EGFP-Plin2 show an ordered recruitment to lipid droplets. (**A**) Lateral views of the anterior intestine in unfed larvae and in larvae at different time-points following the start of feeding with a high-fat meal for 1 hr. Fish were heterozygous for both *Fus(plin3-RFP)* and *Fus(EGFP-plin2)*. Images are representative of three independent experiments (15–25 fish per experiment); data presented are from one experiment. Scale = 50 µm. (**B**) Higher magnification micrographs of lipid droplets highlight the transition from Plin3-RFP to EGFP-Plin2 on the surface of lipid droplets over time after a high-fat meal. Scale = 10 µm. (**C**) Manders' colocalization coefficients for a subset of images was quantified following Costes method for automatic thresholding. Mean ± SD, n = 4 fish per time-point.

The online version of this article includes the following figure supplement(s) for figure 3:

**Source data 1.** Source data associated with *Figure 3C*.

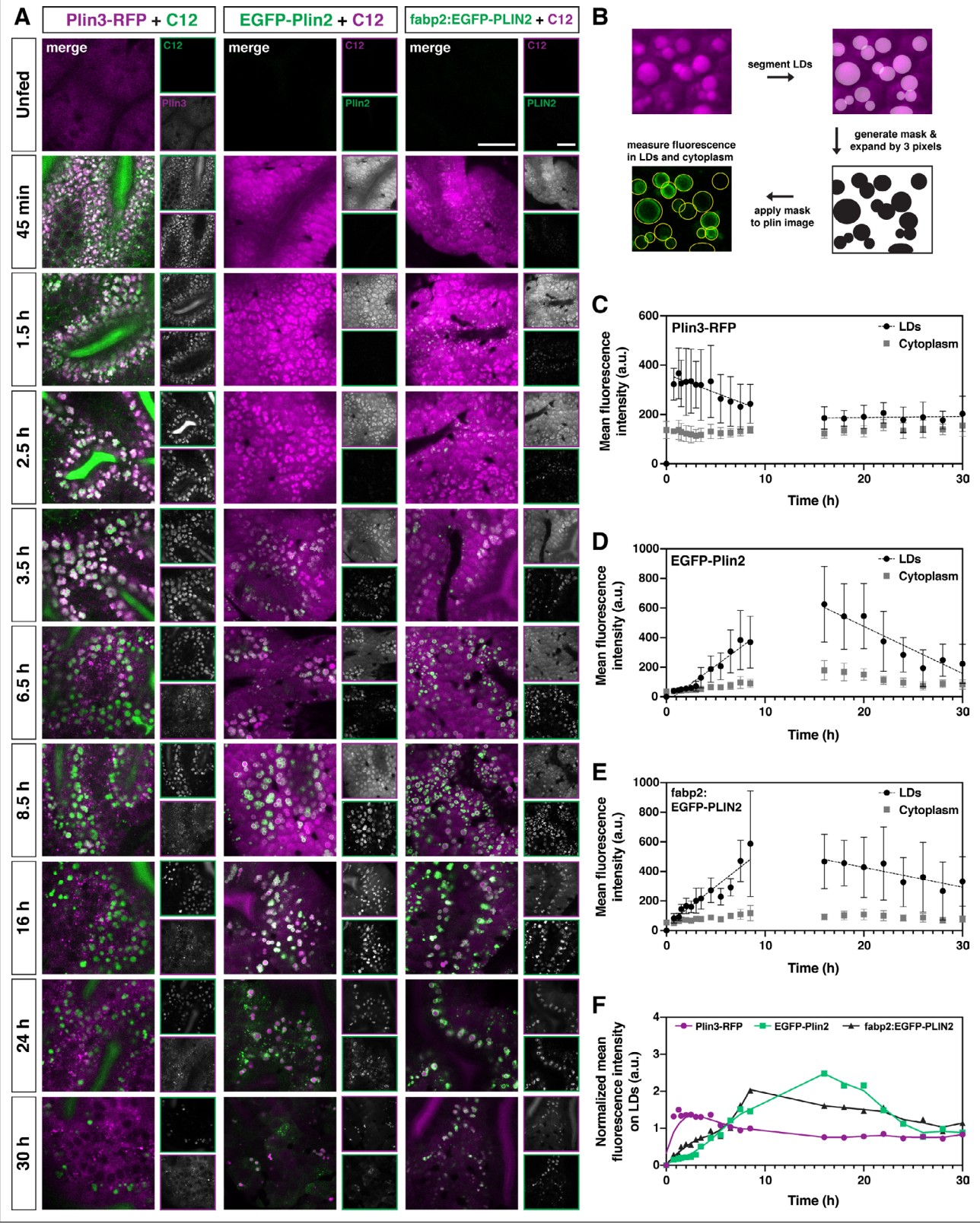

**Figure 4.** Plin3 labels nascent lipid droplets and is replaced by a coat of Plin2, which remains with the droplets for the remainder of their lifetime. (**A**) Representative confocal micrographs of *Fus(plin3-RFP)/+*, *Fus(EGFP-plin2)/+*, or *Tg(fabp2:EGFP-PLIN2)/+* fish fed a high-fat meal containing either green or red BODIPY C12 fatty acid analog. Fish were fed for a total of 90 min and chased at room temperature for 30 hr after the onset of the meal. Scale = 20 μm. See also *Figure 4—video 1*, *Figure 4—video 2*, *Figure 4—video 3*. (**B**) Perilipin fluorescence associated with the lipid droplets and

*Figure 4 continued on next page*

*Figure 4 continued*

in the cytoplasm was assessed by first segmenting the lipid droplets based on the BODIPY C12 signal, the segmented regions were expanded by three pixels in all dimensions (360 nm) to create a mask which was then applied to corresponding Plin fluorescent image. Mean fluorescence intensity was assessed both in the masked regions (LDs) and outside (cytoplasm). (C–E) Mean fluorescence intensity of Plin3-RFP (**C**), EGFP-Plin2 (**D**), and over-expressed human EGFP-PLIN2 (**E**) in the cytoplasm and associated with lipid droplets over time following a high-fat meal (mean ± SD). Data represent two experiments per genotype, each experiment contained larvae from two clutches (n = 9–12 (**C**), 4–18 (**D**), and 3–15 (**E**) fish per time-point). Due to the complexity and length of the time-course, the 0–8.5 and 16–30 hr time-points represent data from different clutches. The linear correlation coefficients (R) for the 0.75–8.5 hr time-points are –0.3266 (y = –15.20 x + 364), 0.7732 (y = 48.91 x + (–34.40)), and 0.7277 (y = 52.33 x + 36.64) for Plin3, Plin2, and PLIN2, respectively. For the 16–30 hr time-points, the R coefficients are 0.0349 (y = 0.3730 x + 180.6), –0.6321 (y = –31.84 x + 1113), and –0.2865 (y = –13.33 x + 693.8) for Plin3, Plin2, and PLIN2, respectively. (**F**) Data from C, D, and E were normalized based on the overall mean fluorescence from all individual data points for each genotype and plots were overlaid to better show the relationship between genotypes. A LOWESS local regression line was applied to the normalized data-sets for visualization purposes only.

The online version of this article includes the following video and figure supplement(s) for figure 4:

**Source data 1.** Source data associated with *Figure 4C–F*.

**Figure supplement 1.** *Tg(fabp2:EGFP-PLIN2)* fish over-express human PLIN2 in the yolk syncytial layer and intestine.

**Figure supplement 1—source data 1.** Source data associated with *Figure 4—figure supplement 1D and E*.

**Figure supplement 2.** Remaining LD area at 30 hr.

**Figure supplement 2—source data 1.** Source data associated with *Figure 4—figure supplement 2*.

**Figure 4—video 1.** Plin3-RFP and BODIPY C12 time series.
https://elifesciences.org/articles/66393/figures#fig4video1

**Figure 4—video 2.** EGFP-Plin2 and BODIPY C12 time series.
https://elifesciences.org/articles/66393/figures#fig4video2

**Figure 4—video 3.** EGFP-PLIN2 and BODIPY C12 time series.
https://elifesciences.org/articles/66393/figures#fig4video3

---

and whether Plin2 could bind to nascent lipid droplets sooner if the protein was already present. To investigate this question, we repeated the intestinal time-course imaging in transgenic fish stably over-expressing human PLIN2 under the control of the zebrafish fatty acid binding protein 2 promoter, *Tg(fabp2:EGFP-PLIN2)*. This promoter drives expression in the yolk syncytial layer and in the intestine (*Her et al., 2003*; *Figure 4—figure supplement 1*). In unfed fish, EGFP-PLIN2 signal cannot be detected above the autofluorescence visible in the intestine at 6 dpf (*Figure 4A* unfed, *Figure 4—figure supplement 1*), likely because any protein produced is rapidly degraded in the absence of lipid droplets (*Xu et al., 2005*). However, we hypothesized that the mRNA would be available immediately for translation and PLIN2 protein would be produced more quickly than in the *Fus(EGFP-plin2)* line. Indeed, we visually detected EGFP-PLIN2 on lipid droplets at 45 min, our earliest time-point after the start of the meal (*Figure 4A and E*) and the difference between fluorescence on lipid droplets vs. in the cytoplasm reached statistical significance by 3.5 hr (p = 0.0134; two-way ANOVA with Šídák multiple comparisons test), suggesting that the droplets are not prohibitive to Plin2 binding at the early times. Although we also hypothesized that over-expression of EGFP-PLIN2 might delay the degradation of lipid droplets in the enterocytes, the fraction of intestine area covered by remaining lipid droplets at 30 hr was not different between the three transgenic lines or wild-type fish (p = 0.2922, Kruskal-Wallis test, n = 9–23 fish, *Figure 4—figure supplement 2*). Furthermore, over-expression of EGFP-PLIN2 in the intestine does not affect the standard length or mass of the fish at 6 months of age (*Figure 4—figure supplement 1*).

After Plin3-RFP is lost from the lipid droplets, some brighter RFP+, BODIPY C12-negative puncta are noted in the cytoplasm (*Figures 3B and 4A*; *Figure 4—video 1*). Similarly, bright EGFP-Plin2 puncta that are not associated with the red BODIPY C12 signal are often visible at the later time-points (24–30 hr) (*Figure 4B*; *Figure 4—video 2*, *Figure 4—video 3*). We hypothesize that these puncta may represent perilipin accumulation in degradative organelles such as lysosomes or autophagosomes, both of which play a role in PLIN and lipid droplet degradation (*Kaushik and Cuervo, 2015*; *Schott et al., 2019*; *Schulze et al., 2020*; *Khaldoun et al., 2014*).

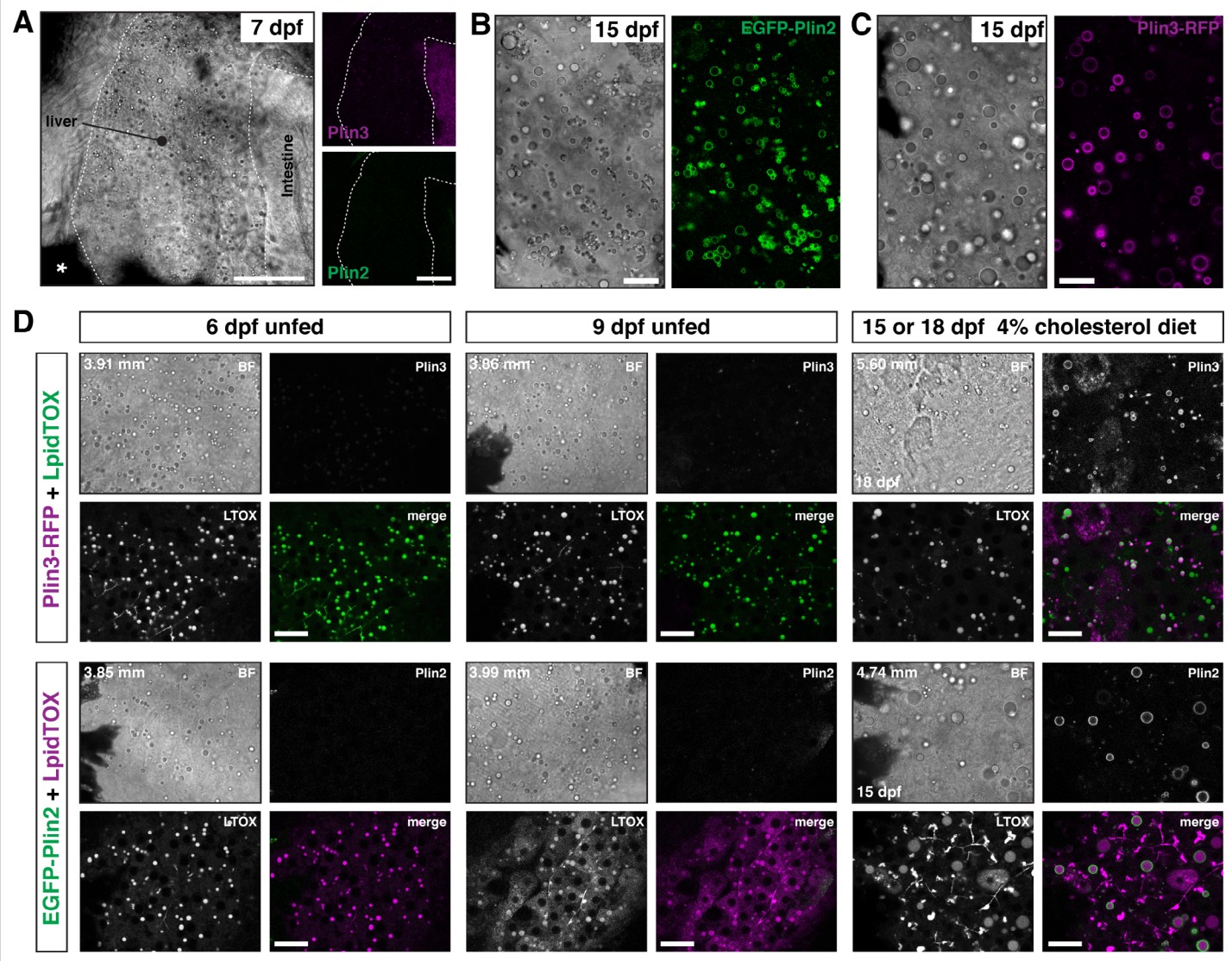

**Figure 5.** EGFP-Plin2 and Plin3-RFP are only expressed in the liver of older larvae. (**A**) Lateral view of the liver in a 7 days post fertilization (dpf) zebrafish larvae heterozygous for both *Fus(plin3-RFP)* and *Fus(EGFP-plin2)*. Scale = 50 μm. (**B, C**) Brightfield and fluorescence liver micrographs from 15 dpf larval zebrafish fed a diet of Gemma +4 % cholesterol for 10 days. Lipid droplets in hepatocytes can be labeled with EGFP-Plin2 (**B**) and with Plin3-RFP (**C**). Scale = 20 μm. (**D**) Liver micrographs from 6, 9, 15 (*Fus(EGFP-plin2)*) or 18 dpf (*Fus(plin3-RFP)*) zebrafish. LipidTox Green or Red (LTOX) labels the hepatic ducts and the lipid droplets, which are also visible in the brightfield image (BF). Where noted, fish were fed a Gemma +4 % cholesterol diet starting at 5 dpf. Standard length of the imaged fish is noted on the upper left corner of each set of image, see source data for additional standard length data. Scale = 20 μm. Images are representative of at least 15 fish from three or more clutches.

The online version of this article includes the following figure supplement(s) for figure 5:

**Source data 1.** Standard length data associated with *Figure 5D*.

## Plin2 and Plin3 decorate hepatic lipid droplets in older larvae

While the intestine absorbs dietary lipid, the vertebrate liver plays an equally important role in whole body lipid metabolism. Specifically, the liver is responsible for de novo lipid synthesis, local lipid storage in cytoplasmic lipid droplets, and lipid export via lipoprotein secretion. Brightfield imaging of livers in larvae at 6–7 dpf suggest they contain lipid droplets (*Figure 5A and D*). These droplets can be labeled with LipidTOX lipophilic dyes (*Figure 5D*) and we have shown previously that these droplets can store lipids containing BODIPY fatty acid analogs (*Carten et al., 2011*; *Quinlivan et al., 2017*). However, in these young larvae, the liver lipid droplets are very rarely labeled with EGFP-Plin2 or Plin3-RFP (*Figure 5A and D*). Continued fasting of these young larvae does not induce expression of either EGFP-Plin2 or Plin3-RFP

despite the continued presence of lipid droplets (*Figure 5D*, 9 dpf unfed). In contrast, hepatic lipid droplets in older larvae are labeled with EGFP-Plin2 and Plin3-RFP (*Figure 5B,C,D*, 15 or 18 dpf). While labeled lipid droplets are present in fish fed our standard Gemma diet (not shown), by supplementing the diet with 4 % cholesterol, the hepatic lipid droplets tend to be more abundant and are more often decorated by Plin2 and Plin3 (*Figure 5B and D*). These data suggest transcriptional activation of *plin2* and *plin3* may be controlled by both developmental and metabolic programs.

## EGFP-Plin2 and Plin3-RFP decorate adipocyte lipid droplets

Using fluorescent lipophilic dyes, Minchin and Rawls have described 34 distinct regions of adipose tissue in zebrafish, including 5 visceral and 22 subcutaneous adipose tissue depots (*Minchin and Rawls, 2017b*; *Minchin and Rawls, 2017a*). The earliest depots to develop are the abdominal visceral adipose tissue (AVAT) and the pancreatic visceral adipose tissue (PVAT), which appear lateral and posterior to the distal swim bladder at ~4.4–5.5 mm standard length (*Minchin and Rawls, 2017a*). Confocal imaging in fish carrying the *plin* knock-in alleles indicates that the lipid droplets in the adipocytes of the PVAT/AVAT tissue are labeled with both EGFP-Plin2 and Plin3-RFP (*Figure 6A–D*). Lipid droplets are easily identified following incubation of the fish in the LipidTOX lipophilic dyes (*Figure 6B and C*) and adipocyte lipid droplets can be simultaneously labeled with both Plin2 and Plin3 (*Figure 6D*). As the fish grow and mature, these adipose depots become larger and the lipid droplets continue to be labeled by Plin2 and Plin3 (*Figure 6F–H*). However, due to the expression of Plin3-RFP throughout the cytoplasm of intestinal enterocytes, the substantial green autofluorescence of the intestinal lumen, and the presence of EGFP-Plin2-labeled lipid droplets in the intestine, it is often difficult to distinguish the fluorescence associated with the Plin proteins in the adipocytes in these depots. We did not note any obvious changes in adipose tissue area in the transgenic fish when compared with their wild-type siblings and the standard length of the fish was not altered (*Figure 6—figure supplement 1*). Thus, we expect that the Plin knock-in alleles will be valuable tools to assist in studies of adipocyte lipid droplet dynamics in vivo during development and in pathological conditions.

## EGFP-Plin2 indicates the presence of lipid droplets around neuromasts

Unexpectedly, when imaging larvae at 15 dpf, we consistently noted small EGFP-Plin2-positive lipid droplets in the neuromasts of the posterior lateral line (*Figure 7*). The neuromasts are sensory epithelial receptor organs that contain hair cells that respond to changes in movement and pressure of the surrounding water (*Chitnis et al., 2012*; *Metcalfe et al., 1985*; *Dijkgraaf, 1963*). The lipid droplets appear at the edge of the organs, suggesting that they are likely not within the hair cells, but may be located in either the support cells, mantle cells (*Chitnis et al., 2012*; *Steiner et al., 2014*; *Hernández et al., 2007*; *McDermott et al., 2010*), or in the neuromast border cells (*Seleit et al., 2017*). These findings are consistent with the report of lipid droplets adjacent to neuromasts in zebrafish imaged with lattice light sheet PAINT microscopy (*Legant et al., 2016*). Crossing the *Fus(EGFP-plin2)* line to transgenic reporter lines for the different cell types (*Parinov et al., 2004*; *Steiner et al., 2014*; *Thomas and Raible, 2019*) will allow the cellular location of these organelles to be identified and may provide insight into the possible role the lipid droplets play in neuromast physiology.

## Additional transgenic lines are also available for over-expression of human PLIN2 and PLIN3

While the knock-in lines are superior for imaging Plin2 and Plin3 in the zebrafish because they are expressed under the control of the endogenous promoter and regulatory elements, we also have a number of additional Tol2-based transgenic lines available which could be useful in specific contexts or for specific purposes. These lines express human PLIN2 or PLIN3 under the control of the zebrafish *fabp2*, *fabp10a,* or *hsp70l* promoters for over-expression in the yolk syncytial layer and intestine, liver, or throughout the larvae following heat shock, respectively (*Table 1* and *Figure 8*). The decision to over-express the human orthologs was mostly out of convenience of already having the clones present in our laboratory and evidence from cell culture studies that fusions of human PLIN proteins with fluorescent proteins were functional (*Targett-Adams et al., 2003*). However, we also appreciated that human proteins are more amenable to detection with commercially available antibodies in downstream applications and might provide an opportunity to maintain tissue-specific or temporal control of PLIN expression when targeting the endogenous loci for gene editing.

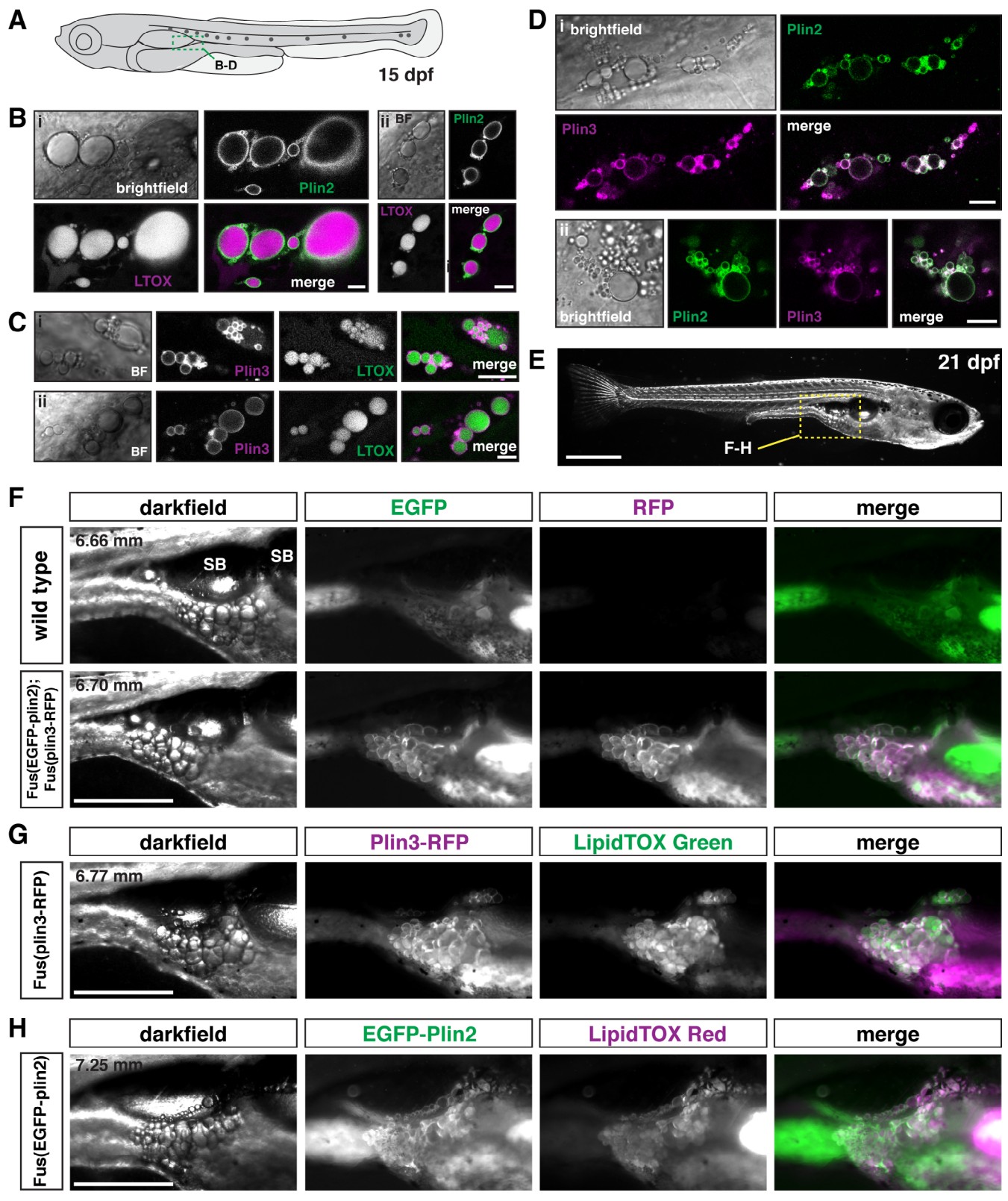

**Figure 6.** EGFP-Plin2 and Plin3-RFP decorate lipid droplets in visceral adipose tissue. (**A**) Cartoon of 15 days of post fertilization (dpf) larval zebrafish showing the general location (green box) of images in panels B–D. (**B, C**) Examples of Plin-positive adipocyte lipid droplets. Fish heterozygous for the noted transgene were fed Gemma for 10 days and then stained with either LipidTOX Red (**B**) or Green (**C**) dyes for a minimum of 2 hr prior to imaging. Scale = 10 μm; standard length of all fish imaged was 5.14 ± 0.26 mm (mean ± SD, n = 47 fish). (**D**) EGFP-Plin2 and Plin3-RFP co-label the surface of

*Figure 6 continued on next page*

*Figure 6 continued*

adipocyte lipid droplets. Fish are heterozygous for each transgene and were fed Gemma + 4 % cholesterol for 10 days prior to imaging. Scale = 10 µm; standard length of similarly fed double-heterozygous fish at 15 dpf, 5.19 ± 0.42 mm (mean ± SD, n = 14 fish). For B–D, images are representative of at least 10 fish per genotype from three independent clutches. (E) Darkfield whole-mount image of a 21 dpf zebrafish, yellow box indicates region of images in panels F–H; scale = 1 mm. (F) Images of pancreatic visceral adipose tissue/abdominal visceral adipose tissue (PVAT/AVAT) adipose depots in wild type and *Fus(EGFP-plin2)/+; Fus(plin3-RFP)/+* fish at 21 dpf. Adipocyte lipid droplets are co-labeled with Plin2 and Plin3. Note the substantial autofluorescence in the EGFP channel in wild-type fish. (G, H) Examples of PVAT/AVAT in either *Fus(plin3-RFP)/+* fish stained with LipidTOX Green (G) or *Fus(EGFP-plin2)/+* stained with LipidTOX Red (H) to more clearly show the relationship of the Plin fluorescence relative to the lipid content of the droplets. For (F–H), fish were fed standard Gemma diet for 15 days and fasted for 24 hr prior to imaging to decrease the fluorescence in the intestine resulting from both EGFP-Plin2 expression and/or LipidTOX-labeled lipid droplets in enterocytes. Scale = 500 µm; standard length of the fish shown is noted in upper left corner of the darkfield images. Images are representative of 18–27 fish from two independent clutches.

The online version of this article includes the following figure supplement(s) for figure 6:

**Source data 1.** Standard length data associated with *Figure 6B–D*.

**Figure supplement 1.** Standard length measurements associated with *Figure 6*.

**Figure supplement 1—source data 1.** Source data associated with *Figure 6—figure supplement 1*.

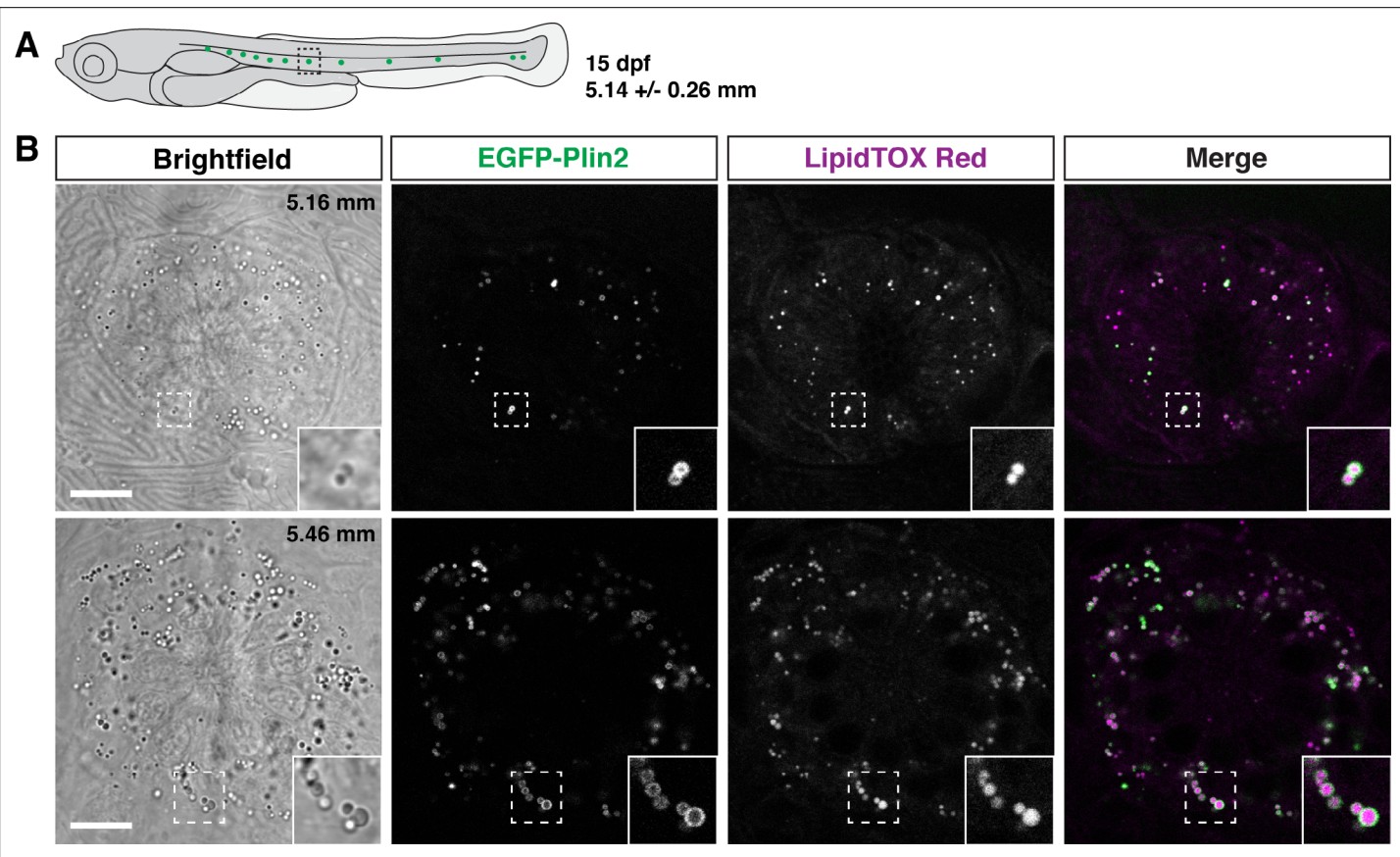

**Figure 7.** EGFP-Plin2 decorates lipid droplets in cells surrounding neuromasts. (A) Cartoon of 15 days of post fertilization (dpf) larval zebrafish showing the general location of lateral line neuromast images shown in panel B. The standard length of larvae imaged at this stage was 5.14 ± 0.26 mm (mean ± SD, n = 47 fish, see *Figure 6—source data 1*). (B) Examples of lipid droplets around neuromasts in fish heterozygous for *Fus(EGFP-plin2)* at 15 dpf. Fish were fed Gemma for 10 days and then stained with LipidTOX Red for a minimum of 2 hr prior to imaging. Insets show enlarged images of the lipid droplets in the boxed regions. Scale = 10 µm, standard length of the fish shown is noted in the brightfield image panel. Images are representative of at least 10 fish from three independent clutches.

**Table 1.** Comparison of available transgenic perilipin lines.

| Transgenic line | Promoter | Coding sequence | Tissue expression |
|---|---|---|---|
| *Fus(EGFP-plin2)* | Integration into the endogenous *plin2* locus | Zebrafish perilipin 2 ENSDART00000175378.2 | Intestine, liver, adipose, neuromasts, rare LDs in yolk syncytial layer |
| *Tg(fabp2:EGFP-PLIN2)* | Zebrafish intestinal fatty acid binding protein (*fabp2*) | Human perilipin 2 ENST00000276914.7 | Yolk syncytial layer, intestine |
| *Tg(fabp10a:EGFP-PLIN2)* | Zebrafish liver fatty acid binding protein (*fabp10a*) | Human perilipin 2 ENST00000276914.7 | Liver |
| *Tg(hsp70I:EGFP-PLIN2)* | Zebrafish heat shock cognate 70-kd protein, like (*hsp70l*) | Human perilipin 2 ENST00000276914.7 | Widespread tissue expression following heat shock; Labeled LDs observed in intestine and liver |
| *Fus(plin3-RFP)* | Integration into the endogenous *plin3* locus | Zebrafish perilipin 3 ENSDART00000100473.5 | Intestine, liver, adipose; Cytoplasmic in addition to LDs |
| *Tg(fabp2:PLIN3-EGFP)* | zebrafish intestinal fatty acid binding protein (*fabp2*) | Human perilipin 3 ENST00000221957.9 | Yolk syncytial layer, intestine; Cytoplasmic in addition to labeled LDs in intestine |
| *Tg(hsp70I:PLIN3-EGFP)* | Zebrafish heat shock cognate 70-kd protein, like (*hsp70l*) | Human perilipin 3 ENST00000221957.9 | Widespread tissue expression following heat shock; often mosaic; Cytoplasmic in addition to labeled LDs in intestine and liver |

## Discussion

Studies on the role of vertebrate PLINs in modulating lipid droplet dynamics have been mostly performed in cell culture or at limited time-points in fixed tissues. While cell-based work has been, and will continue to be, crucial for detailed investigations of lipid droplet biogenesis at the ER, protein structure-function analyses, and the molecular underpinnings of how lipid droplet size and number is regulated, studies in vivo are essential for understanding how PLINs impact whole animal lipid physiology. Additionally, in systems that are difficult to model ex vivo, lipid droplet dynamics in cultured cells may not recapitulate what is observed in vivo. For instance, dietary lipid absorption in the intestinal enterocytes is profoundly influenced by bile (*Hofmann and Borgstroem, 1964*) and microbiota (*Semova et al., 2012*; *Martinez-Guryn et al., 2018*) which are hard to replicate ex vivo. Until very recently, live imaging of PLINs in vivo has been limited to yeast and invertebrate systems. Therefore, the fluorescent knock-in *Fus(EGFP-plin2)* and *Fus(plin3-RFP)* zebrafish reporter lines described here provide important new tools for studying lipid droplets and the in vivo role of Plin2 and Plin3 in a vertebrate system.

We chose to develop these reporters in the zebrafish system because of the ease of live imaging in the optically clear embryonic and larval tissues. Additionally, we engineered the fluorescent protein tags into the endogenous loci to minimize over-expression artifacts. Our initial validation studies indicate that these lines faithfully recapitulate the endogenous tissue expression patterns and expected subcellular localization of Plin2 and Plin3 in the larval zebrafish. Furthermore, as a demonstration of their utility to uncover new biology, we use these tools to describe the ordered recruitment of Plin3 followed by Plin2 onto lipid droplets in the intestine following a high-fat meal. Plin3-RFP, which is present in the cytoplasm of enterocytes in unfed fish, localizes rapidly to nascent lipid droplets, and remains associated for ~6–7 hr until it is once again found primarily in the cytoplasm. In contrast, EGFP-Plin2 only begins to be visible at ~2.5 hr, and it is almost exclusively associated with lipid droplets for the remainder of their lifetime.

One concern with using fluorescently tagged proteins is that the tag alters the folding-time or stability of the protein, which in turn may alter the kinetics of the phenomenon being studied. Because Plin3-RFP is already present in the cytoplasm before the emergence of lipid droplets in the intestine,

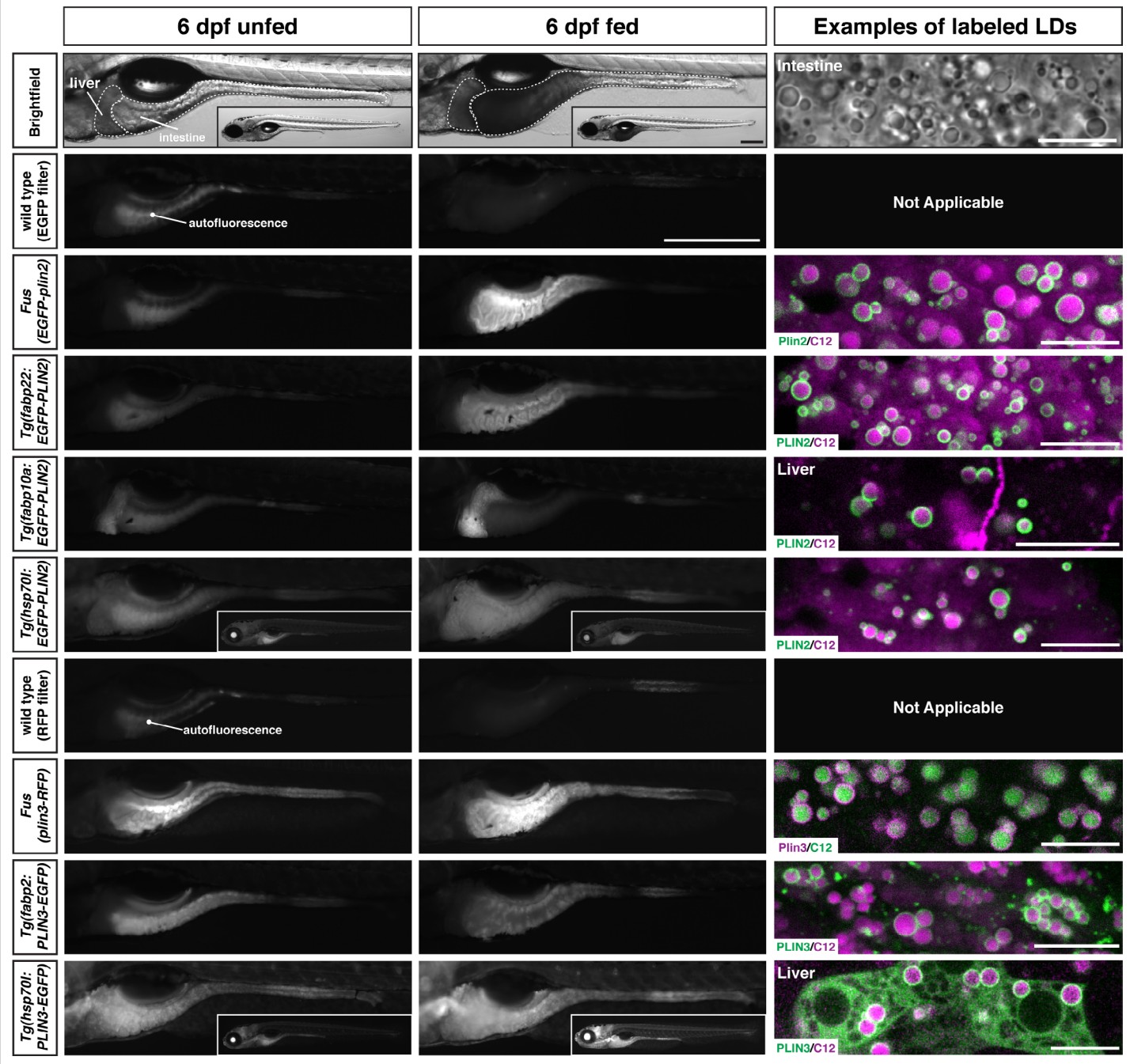

**Figure 8.** Whole-mount images and examples of perilipin-labeled lipid droplets corresponding to the transgenic zebrafish lines noted in *Table 1*. All fish are heterozygous for the noted transgene. Heat shock transgenic lines were incubated at 37 °C for 45 min prior to feeding. For whole-mount images, larvae were fed for 2 hr with a high-fat meal and imaged 3–4.5 (PLIN3 lines) or 5–8 hr (PLIN2 lines) following the start of the feed. Where appropriate, images of whole fish are included as insets. Scale = 500 μm for main images and insets. In the right column, confocal micrographs are included to show the fluorescent perilipin proteins labeling BODIPY-C12-positive lipid droplets in the various transgenic lines following a high-fat meal. Unless noted, images are from the intestine. Scale = 10 μm for each image. Images are representative of at least 10 fish from three independent clutches.

the timing of the appearance of this transgene on lipid droplets is likely similar to the unlabeled Plin3 allele. We suspect that the additional time necessary to translate and fold the EGFP tag (~30–60 min, *Heim et al., 1995*; *Balleza et al., 2018*) probably delays the appearance of EGFP-Plin2 on lipid droplets compared to unlabeled Plin2. However, the visible detection of EGFP-Plin2 starting at ~2–3 hr is

still consistent with the peak in *plin2* mRNA expression in zebrafish gut between 1 and 3 hr following the start of a high-fat meal (*Zeituni et al., 2016*). Importantly, we did not note any differences in the area covered by remaining lipid droplets at 30 hr between *Fus(EGFP-plin2)/+* fish and wild-type siblings, suggesting that the tag did not alter the lifetime of the lipid droplets. Although an N-terminal FLAG tag was reported to decrease ubiquitination and degradation of unbound PLIN2 (*Takahashi et al., 2016*), we did not see accumulations of either zebrafish or human EGFP-PLIN2 in the cytoplasm. Furthermore, fluorescently tagged lipid droplet-associated PLIN2 is permissive to both chaperone-mediated autophagy (*Kaushik and Cuervo, 2015*) and direct transfer into the lysosome (*Schulze et al., 2020*). Therefore, we have little reason to believe the fluorescent tags used in this study are interfering with lipid droplet dynamics; however, we cannot test these hypotheses directly due to the current lack of antibodies that recognize the zebrafish Plin2 and Plin3 orthologs. Despite these caveats, these reporters indicate a clear progression from Plin3 to Plin2 on intestinal lipid droplets following a high-fat meal in zebrafish.

The shift from Plin3 to Plin2 over time is consistent with displacement of PLIN3 by PLIN2 as lipid droplets grow in size in cultured 3T3-L1 adipocytes (*Wolins et al., 2005*) and a recent investigation detailing hierarchical binding for PLINs (*Ajjaji et al., 2019*). However, it contrasts with the finding in mice that PLIN2 is not found on enterocyte lipid droplets 3 hr following a lipid gavage (*Lee et al., 2009*). Therefore, it is unclear whether this ordered recruitment in the intestine is specific to fish, or whether it would also be observed in mammals if more frequent sampling was performed following a high-fat meal. Our finding that over-expressed human EGFP-PLIN2 appears earlier on the lipid droplets than the endogenously expressed EGFP-Plin2 suggests that the sequential targeting of Plin3 followed by Plin2 may simply reflect the relative expression level and affinity of the PLIN proteins in the enterocytes rather than size or maturation of the droplets. Going forward, it will be interesting to determine whether Plin3 is removed earlier from the lipid droplet surface in fish expressing both Plin3-RFP and human EGFP-PLIN2. Furthermore, the physiological significance of this shift in PLIN lipid droplet association in enterocytes remains to be determined, though we hypothesize that it may aid in the regulation of chylomicron production and post-prandial plasma lipid levels.

The PLIN2 deficient knock-out mouse is less efficient at absorbing dietary fat, has higher fecal lipids, and is associated with an altered microbiome, thereby making the knock-out mice resistant to the adverse effects of a high-fat diet (*Frank et al., 2015*). However, the precise cell biological processes that are altered in the enterocytes of these mice remain to be described. In future studies, imaging Plin3-RFP in Plin2-null zebrafish will allow us to assess whether Plin3 remains on lipid droplets when Plin2 is absent and determine whether the loss of Plin2 alters the lifetime of the intestinal lipid droplets following a high-fat meal and whether shorter-lived lipid droplets influence the phenotypes described in the mouse model. In 2012, we partnered with the Rawls Laboratory to show that specific microbes are associated with profound increases in intestinal lipid droplet number and/or size (*Semova et al., 2012*), work recently confirmed in mice (*Martinez-Guryn et al., 2018*); however, we still do not understand how the microbes accomplish this feat. Using the Plin reporter lines in the context of different microbial communities in live fish will allow us to test whether certain microbes differentially alter Plin expression or lipid droplet dynamics in enterocytes.

As the fish mature, the utility of these Plin knock-in transgenic lines becomes evident in tissues beyond the intestine. Not surprisingly, both the Plin2 and Plin3 reporters label lipid droplets in the liver and adipose tissue. By 5 dpf, the zebrafish liver has differentiated hepatocytes, and functional biliary and vasculature systems (*Wilkins and Pack, 2013*). It performs many hepatic functions, including bile secretion (*Farber et al., 2001*), insulin responsiveness (*Toyoshima et al., 2008*), storage of lipids and glycogen (*Howarth et al., 2013*), as well as secretion of serum proteins such as vitamin D binding protein (*Noël et al., 2010*) and transferrin (*Mudumana et al., 2004*), suggesting that it is functionally mature. Thus, the delay in appearance of Plin expression in the liver until late larval stages was unexpected given that we observed LipidTOX-labeled lipid droplets at 6 dpf. This finding suggests that aspects of hepatic lipid metabolism may not reach full maturity until later in development. Furthermore, while PLIN1 is primarily found in adipose tissue in mammals (*Greenberg et al., 1991*), mRNA expression levels of *plin1* in the livers of adult grass carp are even higher than in adipose tissue (*Huang et al., 2020*), suggesting that Plin1 might be coating the hepatocyte lipid droplets in these young zebrafish larvae. Using these new transgenic lines to study the differential expression of perilipins in hepatocytes in young vs. older larvae provides an opportunity to discover novel insights into the

transcriptional regulation of PLINs and how the PLINs influence hepatic lipid storage and mobilization in vivo.

At 15 dpf (standard length ~5 mm), small clusters of visceral adipocytes containing lipid droplets co-labeled with EGFP-Plin2 and Plin3-RFP were visible in the larval zebrafish, consistent with previous characterization of zebrafish adipose tissues using lipophilic dyes (*Minchin and Rawls, 2017b*; *Minchin and Rawls, 2017a*). As these adipose depots expand, Plin2 and Plin3 continue to decorate the large unilocular lipid droplets at 21 dpf (~7 mm standard length). Going forward, it will be interesting to use these transgenic lines to determine whether Plin2 and Plin3 are eventually displaced by Plin1 on the adipocyte lipid droplets and if so, whether this correlates with the size of the droplets or perhaps some other marker of droplet maturation.

As these knock-in reporter lines are used to further characterize lipid metabolism in different physiological and pathological conditions, we expect that Plin2- and Plin3-labeled lipid droplets will be identified in other zebrafish tissues. For example, in mammals, lipid droplets and PLINs are present in muscle tissue (*Dalen et al., 2007*; *Wolins et al., 2006*; *Yamaguchi et al., 2006*), and muscle can profoundly impact whole animal nutrient physiology in both normal and a variety of disease states (*Gemmink et al., 2020*). If found to be present in muscle, these Plin reporter lines could enable zebrafish studies on the effect of diet, genes, and exercise on intramyocellular lipid. Moreover, our discovery of Plin2-positive lipid droplets in lateral line neuromasts suggests that perilipins may be identified in tissues and cells beyond those that are typically known to harbor cytoplasmic lipid droplets.

In humans, like in zebrafish, PLIN2 is expressed in the liver and intestine, organs that play a major role in lipoprotein metabolism. The human gene region that includes PLIN2 is strongly associated with plasma cholesterol levels (p = 1.362e-20; Type 2 Diabetes Knowledge Portal. type2diabetesgenetics.org. 2021 July 26; https://t2d.hugeamp.org/region.html?chr=9&end=19199288&phenotype=LDL&start=19058373) and a PLIN2 mutation (Ser251Pro) is associated with decreased atherogenic lipids (plasma triglyceride and very low-density lipoprotein) (*Magné et al., 2013*). While there remains a question as to whether Ser251Pro causes the phenotype (*Sentinelli et al., 2016*), taken together these human genome-wide association data suggest that PLIN2 can impact the development of metabolic and cardiovascular diseases although a cell biological mechanism for these phenotypes remains to be described. Using these zebrafish Plin reporter lines in the context of diets (*Stoletov et al., 2009*; *Turola et al., 2015*; *Sapp et al., 2014*) and established zebrafish mutations and disease models (*Hölttä-Vuori et al., 2013*; *Maddison et al., 2015*; *Liu et al., 2015*; *Liu et al., 2018*; *O'Hare et al., 2014*) may provide mechanistic insights that connect PLIN cell biology to metabolic and cardiovascular diseases.

In summary, the *Fus(EGFP-plin2)* and *Fus(plin3-RFP)* knock-in zebrafish lines provide the opportunity to study PLINs and lipid droplet biology in vivo at the organelle, cell, tissue, organ, and whole animal level. These lines exploit the advantages of the zebrafish model and will be important tools to understand how lipid droplet dynamics are affected by different genetic and physiological manipulations.

## Materials and methods
### Zebrafish husbandry and maintenance

All procedures using zebrafish (*Danio rerio*) were approved by the Carnegie Institution Department of Embryology Animal Care and Use Committee (Protocol #139). Zebrafish stocks (AB line) were maintained at 27 °C in a circulating aquarium facility with a 14:10 hr light:dark cycle. For propagation and stock maintenance, starting at 5.5 dpf, larvae were fed with GEMMA Micro 75 (Skretting) 3 × a day until 14 dpf, GEMMA Micro 150 3 × a day+ Artemia 1 × daily from 15 to 42 dpf and then GEMMA Micro 500 1 × daily supplemented once a week with Artemia. The nutritional content of GEMMA Micro is: 59 % protein; 14 % lipids; 0.2 % fiber; 14 % ash; 1.3 % phosphorus; 1.5 % calcium; 0.7 % sodium; 23,000 IU/kg vitamin A; 2800 IU/kg vitamin D3; 1000 mg/kg vitamin C; 400 mg/kg vitamin E. Embryos were obtained by natural spawning and raised in embryo medium at 28.5 °C in culture dishes in an incubator with a 14:10 hr light:dark cycle. Zebrafish sex is not determined until the juvenile stage, so sex is not a variable in experiments with embryos and larvae.

## High-fat and high-cholesterol diets

To feed 6 dpf larvae a high-fat meal, larvae were immersed in a solution of 5 % chicken egg yolk liposomes in embryo media for 1–2 hr on an orbital shaker at 29 °C (for detailed protocol, see *Zeituni and Farber, 2016*). Where noted, BODIPY (558/568)-C12 (D3835, Thermo Fisher Scientific) or BODIPY FL-C12 (D3822, Thermo Fisher Scientific) were included in the egg yolk solution at 4 µg/ml. Following feeding, larvae were washed in embryo media and screened for full guts by examining intestinal opacity under a stereomicroscope. Fed larvae were either maintained in embryo media until imaging, fixed immediately for in situ hybridization, or guts were extracted and frozen for qRT-PCR. A high-cholesterol diet was made by soaking Gemma Micro 75 in a diethyl ether and cholesterol (Sigma-Aldrich C8667) for a final content of 4% w/w cholesterol after ether evaporation (based on *Stoletov et al., 2009*). Larvae were fed with this high-cholesterol diet 3 × daily from 5.5 to 15 dpf where noted.

## LipidTOX staining

To label the neutral lipids stored in lipid droplets in tissues other than the intestine and/or in the absence of feeding, fish were stained with either HCS LipidTOX Green (Thermo Fisher Scientific, H34475) or HCS LipidTOX Red (Thermo Fisher Scientific, H34476) neutral lipid stains. Fish were incubated with LipidTOX dyes at 1:5000 dilution in either embryo media (6, 9 dpf) or system water (15, 21 dpf) in six-well dishes (6, 9, 15 dpf) or 10 cm dishes (21 dpf) for a minimum of 2 hr prior to imaging.

## Whole-mount in situ hybridization

Zebrafish embryos were staged according to *Kimmel et al., 1995* and fixed with 4 % paraformaldehyde in phosphate buffered saline overnight at 4 °C, washed twice with MeOH and stored in MeOH at –20 °C. To generate riboprobes, 754 bp of the *perilipin 2* (*plin2; ENSDARG00000042332; ENSDART00000175378.2* transcript) and 900 bp of the zgc:77,486 (*perilipin 3; plin3;* ENSDARG00000013711, *ENSDART00000100473.5* transcript) (GRCz11) mRNA sequences were amplified from cDNA using the primers noted in *Supplementary file 1* and TA cloned into dual promoter pCRII-TOPO (Thermo Fisher Scientific, K207020). Sense and anti-sense probes were synthesized using the DIG RNA labeling kit (Roche 11277073910) using T7 and SP6 polymerases (Roche 10881767001 and 10810274001). Whole-mount in situ hybridization was performed as previously described (*Thisse and Thisse, 2008*) on 6 dpf unfed and high-fat fed larvae. Larvae were mounted in glycerol and imaged using a Nikon SMZ1500 microscope with HR Plan Apo 1 × WD 54 objective, Infinity 3 Lumenera camera, and Infinity Analyze 6.5 software or a Nikon E800 microscope with a 20 X/0.75 Plan Apo Nikon objective and Canon EOS T3 camera using EOS Utility image acquisition software.

## DNA extraction and genotyping

Genomic DNA was extracted from embryos, larvae, or adult fin clips using a modified version of the HotSHOT DNA extraction protocol (*Meeker et al., 2007*). Embryos or tissues were heated to 95 °C for 18 min in 100 µl of 50 mM NaOH, cooled to 25 °C, and neutralized with 10 µl of 1 M Tris-HCL pH 8.0. The gDNA extractions and PCR verifying integration of the fluorescent tags into the genomic loci was performed using the REDExtract-N-Amp Tissue PCR kit (Sigma-Aldrich, XNAT). PCR amplicons were run on 1% or 2 % agarose gels in TBE and gels were imaged with Bio-Rad Gel ChemiDoc XRS system and Quantity One software. For primer information, see *Supplementary file 1*.

## RNA isolation, cDNA synthesis, and quantitative RT-PCR

Following a 90 min feed with 5 % chicken egg yolk, guts were dissected from larvae (6 dpf, 10 guts pooled per sample) and stored in RNA*later* (Thermo Fisher Scientific AM7020) at 4 °C for 1 week. RNA was isolated using a Trizol-based RNA prep adapted from *Macedo and Ferreira, 2014*. Samples were subsequently treated with DNase I and purified using the RNA Clean and Concentrator kit (Zymo Research R1013). cDNA was synthesized using the iScript cDNA Synthesis Kit (Bio-Rad Laboratories, Inc, 1708891). qRT-PCR samples were prepared using SsoAdvanced Universal SYBR Green Supermix (Bio-Rad Laboratories, Inc, 1725271). Primers targeting zebrafish *plin2* transcripts were previously validated (see *Supplementary file 1*; *Zeituni et al., 2016*) and zebrafish 18 S (*rps18*) was used as the reference gene (*Otis et al., 2015*). qRT-PCR was performed in triplicate for each sample with the Bio-Rad CFX96 Real-Time System with 45 cycles: 95 °C for 15 s, 59 °C for 20 s, and 72 °C for 20 s.

Results were analyzed with the Bio-Rad CFX Manager 3.0 software and relative gene expression was calculated using the ΔΔCT method (*Livak and Schmittgen, 2001*).

## Genome editing to create PLIN fusion lines

*Fus(EGFP-plin2)* and *Fus(plin3-RFP)* lines were created with TALEN-mediated genome editing, using (*Shin et al., 2014*) as a guide. The genomic region around the location targeted for editing in the *plin2* and *plin3* genes was amplified by PCR and sequenced from multiple wild-type AB fish to identify any discrepancies between the published sequences and Farber Laboratory stocks. During this process, we identified a variable 54 bp region prior to exon 1 in the *plin2-203* transcript (*ENSDART00000175378.2*) (see *Figure 1—figure supplement 1*) and discovered a polymorphism (T > C) in the ATG designated as the start codon in the *plin2-202 ENSDART00000129407.4* transcript. We designed our editing strategy to fuse the EGFP coding sequence in-frame with the *ENSDART00000175378.2* transcript in fish carrying the 54 bp intronic sequence and performed editing only in fish carrying this full-length sequence. Two pairs of TALENs were designed per gene using the Mojo Hand design tool (*Neff et al., 2013*) and cloned with the FusX assembly system and the pKT3Ts-goldyTALEN vector (*Welker et al., 2016*; *Ma et al., 2013*; *Ma et al., 2016*). The designed TALEN pairs for *plin2* (pair 1 TTTCTGCTAACA TGG and AAATAACCAGGTTTGCC; pair 2 TTTCTGCTAACATGGGT and AATAACCAGGTTTGCC) flank a Fok1 restriction site just downstream of the endogenous start codon. The designed TALEN pairs for *plin3* (pair 1 TTGCGCCTCAGATAAC and AATTGCCACACAACCT; pair 2 CAGATAACAGAG AAA and CACACAACCTAAATA) flank a Hph1 site immediately upstream of the endogenous termination codon. TALEN mRNA was in vitro transcribed using the T3 Message Machine Kit (Thermo Fisher Scientific, AM1348), injected into one-cell stage zebrafish embryos, and cutting efficiency of each pair was assessed by monitoring the loss of either FokI (NEB R0109) or HphI (NEB R0158) digestion due to TALEN nuclease activity. Nuclease activity was higher for *plin2* TALEN pair 1 and *plin3* TALEN pair 1 and these were used subsequently for genome integration. Donor plasmids used as templates for homology directed repair were assembled using the three-fragment MultiSite gateway assembly system (Invitrogen, 12537–023). For *plin2*, the 5' element consisted of 594 bp of genomic sequence upstream of th *plin2* start codon, the middle-entry element contained the *plin2* Kozak sequence followed by the EGFP coding sequence lacking a termination codon, and then a linker region coding for three glycine residues that was in-frame with the 3' element which consisted of 900 bp of genomic sequence including and downstream of *plin2* start codon. For *plin3*, the 5' element consisted of the 679 bp of genomic sequence immediately upstream of the termination codon, a middle-entry element of in-frame tagRFP-t (amplified from Addgene #61390, which has been codon modified for zebrafish; *Horstick et al., 2015*) with a C-terminal termination codon, and a 3' element consisting of the 444 bp genomic sequence downstream of the *plin3* termination codon. Genome integration was accomplished by co-injection of 150 pg of TALEN mRNA and 100 pg of donor plasmid into one-cell stage embryos. Injected embryos were raised to adulthood, out-crossed to wild-type fish, and resulting F1 progeny were screened for either EGFP or RFP fluorescence; it was necessary to feed the *Fus(EGFP-plin2)* fish with a high-fat meal in order to detect EGFP-Plin2 fluorescence when integrated correctly. Correct in-frame integration of the fluorescent reporters was confirmed by PCR and sequencing. Following verification, *Fus(EGFP-plin2)* fish were genotyped using primers for EGFP. For primer information, see *Supplementary file 1*.

## Generation of additional transgenic zebrafish

Additional transgenic zebrafish expressing human *PLIN2* (*ENSG00000147872 GRCh38.p13*) or *PLIN3* (*ENSG00000105355*) under the control of various promoters were generated with the Tol2-Gateway molecular cloning system (*Kwan et al., 2007*). The coding sequence of human *PLIN2* with an N-terminal EGFP tag was provided by John McLauchlan (*Targett-Adams et al., 2003*) and re-cloned into pCR8 (Thermo Fisher Scientific, K250020). The human *PLIN3* (TIP47) coding sequence was obtained from Flexgene clones collection (Harvard Medical School, clone ID: HsCD00004695) and re-cloned into pCR8. The intestine-specific intestinal fatty acid binding protein (*fapb2*, p5E –2.3 k fabp2) promoter was provided by Michel Bagnat (*Park et al., 2019*); the liver-specific liver fatty acid binding protein 10a (p5E *fabp10a* [–2.8 kb]), originally described in *Her et al., 2003*, was provided by Brian Link. The heat shock cognate 70 kDa protein, like (*hsp70l*) promoter (p5E-hsp70l), p3E-EGFPpA, pME-EGFP no stop, and p3E-polyA plasmids were originally provided by Chi-bin

Chien (*Kwan et al., 2007*). Gateway recombination was used to combine entry plasmids into the pDestTol2Pa2 plasmid to create *Tg(fabp2:EGFP-PLIN2)*, *Tg(fabp10a:EGFP-PLIN2)*, *Tg(hsp70l:EGFP-PLIN2)*, *Tg(fabp2:PLIN3-EGFP)*, and *Tg(hsp70l:PLIN3-EGFP)* transgene constructs. Plasmids were injected (25–50 pg) along with 40 pg tol2 transposase mRNA into one-cell stage AB embryos. Zebrafish were raised to adulthood, out-crossed to wild-type fish, and resulting embryos were screened for progeny stably expressing the fluorescent constructs. Transgenic larvae expressing a heat shock-inducible construct were incubated at 37 °C for 45 min in 15 ml of embryo media and screened a few hours later. Embryos expressing a *fabp2*-driven construct were screened at 2–4 dpf for EGFP expression in the yolk syncytial layer and embryos expressing the *fabp10a*-driven construct were screened at 5 or 6 dpf following liver development. At least two stable lines per construct were initially generated, the pattern of expression was verified to be the same in each line and subsequently, a single line for each construct was used for experiments and propagated by out-crossing to wild-type AB fish.

## Fluorescence microscopy

For whole-mount imaging, zebrafish larvae were anesthetized with tricaine (Sigma-Aldrich A5040) and mounted in 3 % methylcellulose in embryo media on glass slides and imaged live with a Zeiss Axio Zoom V16 microscope equipped with a Zeiss PlanNeoFluar Z 1 ×/0.25 FWD 56 mm objective, AxioCam MRm camera, EGFP, Cy3, and mCherry filters and Zen 2.5 Blue edition software. The EGFP filter was used for both EGFP and LipidTOX Green imaging, the Cy3 filter was used for Plin3-RFP imaging, and the mCherry filter was used for LipidTOX Red imaging. For confocal imaging of lipid droplets in the tissues of live larvae, fish were anesthetized with tricaine and mounted in 3 % methylcellulose on glass slides with bridged coverslips. Images were obtained with a Leica DMI6000 inverted microscope and Leica 63 ×/1.4 HCX PL Apo oil-immersion objective with a Leica TCS-SP5 II confocal scanner with photomultiplier detectors using Leica Application Suite Advanced Fluorescence 2.7.3.9723 image acquisition software. Images were obtained using four-line averaging and recorded with 12-bit dynamic range. EGFP and BODIPY FL-C12 were excited with an argon laser (488 nm) and had a collection window of 498–530 nm. BODIPY (558/568)-C12 was imaged with 561 laser and collection window of 571–610 nm and mTagRFP-t and LipidTOX Red were imaged with 561 laser and collection window of 575–650 nm. When using green BODIPY FL-C12 or LipidTOX dyes, imaging was performed sequentially for red and green channels.

## Whole-mount intestine PLIN time-course imaging and analysis

To investigate changes in Plin3-RFP and EGFP-Plin2 fluorescence in the whole intestine following a high-fat meal, larvae were fed with a 5 % egg yolk solution for 90 min as described above. Whole-mount images of the intestine were obtained every hour for the first 8 hr and subsequently every 2 hr thereafter for a total of 30 hr from the onset of the feed. Following the meal, all larvae were maintained at room temperature for the duration of the experiment. Fish were collected for *Fus(EGFP-plin2)* genotyping at the 0–5 and 24–30 hr time-points when the EGFP fluorescence is absent or very dim. Fluorescence intensity (raw integrated density) of all images was quantified in Fiji (ImageJ V2.1.0, National Institutes of Health [NIH], Bethesda, MD; *Schindelin et al., 2012*). Mean gray value of the intestine in the corresponding brightfield images from *Fus(EGFP-plin2)*/+ fish at 10 hr was also quantified to obtain an estimate of the amount of lipid consumed.

## Intestinal EGFP-Plin2 and Plin3-RFP confocal time-course imaging and colocalization analysis

To study changes in Plin3-RFP and EGFP-Plin2 localization in the enterocytes at the tissue, cell, and subcellular level, *Fus(plin3-RFP)*/+; *Fus(EGFP-plin2)*/+ larvae were fed for 60 min with 5 % egg yolk in embryo media as described above. Confocal images of the anterior region of the intestine were obtained every hour starting at 30 min and continuing until ~8 hr (63 ×, 1 ×, and 2 × zoom). Costes autothresholding and Manders' colocalization analysis of EGFP and RFP fluorescence was performed on a subset of the 2 × zoom, 8-bit images in Fiji. A region-of-interest was used for each pair of images to ensure the analysis was only being performed in the intestine.

## Subcellular intestine BODIPY C12 and PLIN time-course imaging and analysis

To assess subcellular localization of Plin3-RFP and EGFP-Plin2 proteins in relation to lipid droplets in the intestinal enterocytes over time following a high-fat meal, larvae were fed for 60 min with a 5 % egg yolk solution containing either green BODIPY FL-C12 or red BODIPY (558/568)-C12 at 4 µg/ml as described above. Confocal images of the anterior region of the intestine were obtained approximately every 30 min for the first 3.5 hr, every hour from 4.5 to 8.5 hr, and every 2 hr from 16 to 30 hr (63× + 2 × zoom). Following the meal, all larvae were maintained at room temperature for the duration of the experiment. While *Fus(plin3-RFP)/+* fish were pre-sorted for RFP prior to feeding and imaging, larvae from the *Fus(EGFP-plin2)/+* × AB crosses and *Tg(fabp2:EGFP-PLIN2)/+* × AB crosses were collected for genotyping after imaging to identify wild-type and transgenic fish at the early time-points when EGFP-Plin2 is absent. To quantify the RFP-Plin2 and EGFP-Plin2 fluorescence present in the cytoplasm vs. associated with the BODIPY C12-labeled lipid droplets, we first segmented the BODIPY C12 lipid droplets automatically using Ilastik V1.3.3post3 (*Berg et al., 2019*) (https://ilastik.org). The pixel classifier was trained on example images from different time-points using color/intensity features ranging from σ 1.6 to 5.0 pixels and edge and texture features ranging from σ 0.07 to 5.0 pixels. This classification resulted in a probability map of the brighter BODIPY puncta that represent lipid droplets, vs. the BODIPY C12 signal in the intestinal lumen and cytoplasm of enterocytes. Training and segmentation were performed separately on green BODIPY FL-C12 images and red BODIPY 558/568-C12 images. Following training, images were batch-analyzed and probability maps were exported as 'simple segmentation' files in .tiff format. These files were opened in Fiji and the threshold was adjusted so the image contained only the lipid droplet segmentation as a binary image. Because the lipid droplets are often tightly clustered, it is difficult to segment them into single objects. Therefore, all the lipid droplets were unified into one object mask and this mask was expanded by three pixels (or the equivalent of 360 nm) using the Process> Binary > Dilate function in Fiji. This expansion was necessary to ensure that the mask would encompass the rim of the lipid droplets where the majority of the Plin fluorescent signal is located, especially on larger lipid droplets. This expanded lipid droplet mask was applied to the corresponding fluorescent Plin3-RFP or EGFP-Plin2 image in Fiji and the mean fluorescence intensity in the mask was quantified in addition to the total mean fluorescence intensity without the mask. Because some images included regions of liver or other organs, when necessary, an additional region of interest delineating the intestine was combined with the lipid droplet mask to ensure only the intestine was analyzed. The mean fluorescence in the cytoplasm was obtained by subtracting the fluorescence associated with the lipid droplets from the total fluorescence. In addition, individual data points were also transformed by the mean of all data points (per genotype) to better compare the relative changes in Plin3-RFP, zebrafish EGFP-Plin2, and human EGFP-PLIN2 fluorescence over time. Data from 0.75 to 8.5 or 16–30 hr for each data set was fit using linear regression with the least squares method to determine the coefficient of correlation, R.

## Statistics

All statistical analyses were performed using GraphPad Prism 9 (GraphPad, San Diego, CA). Data are presented as mean ± standard deviation (SD). Statistical tests used are noted in figure legends or text. All experiments were performed with two to three independent replicates (N). The sample size (n) for each experiment is noted in the figure legends.

## Additional software

Graphing was performed with GraphPad Prism (GraphPad Software). DNA, mRNA, and protein sequence alignments were performed with MacVector V15.5 (MacVector, Inc). Micrographs were adjusted and cropped as needed in Fiji (NIH) and figures were assembled in Adobe Illustrator CS5 (Adobe Systems). Microsoft Word and Excel were used for manuscript preparation and data analysis, and references were compiled with EndNote x8 .

## Resource availability

All transgenic zebrafish lines are available upon request. The *Fus(EGFP-plin2)* and *Fus(plin3-RFP)* zebrafish lines are available at the Zebrafish International Resource Center (ZIRC): plin2[c847Tg] (ZIRC Catalog ID: ZL14605) and plin3[875Tg] (ZIRC Catalog ID: ZL14606).

## Acknowledgements

We gratefully acknowledge Andrew Rock, Carmen Tull, Julia Baer, Mackenzie Klemek, and Hannah Kozan for fish husbandry, Amy Kowalski, Lamia Wahba, Blake Caldwell, James Thierer, and Erin Zeituni for synthesis of various Tol2 entry plasmids, Cassandra Bullard and Camden Daby for assembly of TALEN plasmids, Stephanie Yan for technical assistance, Matthew Bray for support with Fiji, Mahmud Siddiqi for advice regarding image analysis, Tabea Moll and Jennifer Anderson for help editing the manuscript, and David Raible for input regarding neuromasts.

## Additional information

### Competing interests

Stephen C Ekker: Reviewing editor, *eLife*. The other authors declare that no competing interests exist.

### Funding

| Funder | Grant reference number | Author |
|---|---|---|
| National Institutes of Health | R01 DK093399 | Steven A Farber |
| National Institutes of Health | R01 GM63904 | Stephen C Ekker<br>Steven A Farber |
| National Institutes of Health | F32DK109592 | Meredith H Wilson |
| Mathers Foundation | | Steven A Farber |

The funders had no role in study design, data collection and interpretation, or the decision to submit the work for publication.

### Author contributions

Meredith H Wilson, Conceptualization, Formal analysis, Funding acquisition, Investigation, Methodology, Resources, Visualization, Writing - original draft, Writing – review and editing; Stephen C Ekker, Funding acquisition, Methodology, Resources; Steven A Farber, Conceptualization, Funding acquisition, Project administration, Resources, Supervision, Writing – review and editing

### Author ORCIDs

Meredith H Wilson http://orcid.org/0000-0002-6152-7127
Stephen C Ekker http://orcid.org/0000-0003-0726-4212
Steven A Farber http://orcid.org/0000-0002-8037-7312

### Ethics

All procedures using zebrafish (Danio rerio) were approved by the Carnegie Institution Department of Embryology Animal Care and Use Committee (Protocol #139).

### Decision letter and Author response

Decision letter https://doi.org/10.7554/eLife.66393.sa1
Author response https://doi.org/10.7554/eLife.66393.sa2

## Additional files

### Supplementary files

- Supplementary file 1. Primers used in this study.
- Transparent reporting form

Data availability

All data generated during this study are included in the manuscript and supporting files; source data files have been provided for Figures 1—figure supplement 1, Figure 2, Figure 3, Figure 4, Figure 4—figure supplement 1 & 2, Figure 5, Figure 6 and Figure 6—figure supplement 1.

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

## Appendix 1

### Appendix 1—key resources table

| Reagent type (species) or resource | Designation | Source or reference | Identifiers | Additional information |
|---|---|---|---|---|
| gene (*Danio rerio*) | plin2 | Ensembl GRCz11 | ENSDARG00000042332; plin2-203 ENSDART00000175378.2 | |
| gene (*Danio rerio*) | zgc:77,486(plin3) | Ensembl GRCz11 | ENSDARG00000013711; zgc:77,486 ENSDART00000100473.5 | |
| Gene (*Homo sapiens*) | PLIN2 | Ensembl GRCh38.p13 | ENSG00000147872; PLIN2-201 ENST00000276914.7 | |
| Gene (*Homo sapiens*) | PLIN3 | Ensembl GRCh38.p13 | ENSG00000105355; PLIN3-201 PLIN3-201 ENST00000221957.9 | |
| Strain, strain background (*Danio rerio*) | *AB* | Zfin | ZDB-GENO-960809–7 | |
| Strain, strain background (*Danio rerio*) | *Fus(EGFP-plin2)* | This paper | | Available on request from Steven Farber |
| Strain, strain background (*Danio rerio*) | *Fus(plin3-RFP)* | This paper | | Available on request from Steven Farber |
| Strain, strain background (*Danio rerio*) | *Tg(fabp2:EGFP-PLIN2)* | This paper | | Zebrafish fabp2 promoter, human PLIN2 coding sequence; Available on request from Steven Farber |
| Strain, strain background (*Danio rerio*) | *Tg(fabp10a: EGFP-PLIN2)* | This paper | | Zebrafish fabp10a promoter, human PLIN2 coding sequence; Available on request from Steven Farber |
| Strain, strain background (*Danio rerio*) | *Tg(hsp70l: EGFP-PLIN2)* | This paper | | Zebrafish hsp70l promoter, human PLIN2 coding sequence; Available on request from Steven Farber |
| Strain, strain background (*Danio rerio*) | *Tg(fabp2:PLIN3-EGFP)* | This paper | | Zebrafish fabp2 promoter, human PLIN3 coding sequence; Available on request from Steven Farber |
| Strain, strain background (*Danio rerio*) | *Tg(hsp70: PLIN3-EGFP)* | This paper | | Zebrafish hsp70l promoter, human PLIN3 coding sequence; Available on request from Steven Farber |
| Recombinant DNA reagent | pCRII(plin3_900)(plasmid) | This paper | | Zebrafish plin3 (zgc:77486) cDNA clone for in situ probe; Available on request from Steven Farber |
| Recombinant DNA reagent | pCRII(plin2)(plasmid) | This paper | | Zebrafish plin2 cDNA clone for in situ probe; Available on request from Steven Farber |
| Recombinant DNA reagent | Pk-GoldyTal(plin2 Pair 1 LeftTAL 1)(plasmid) | This paper | | TALEN targets: TTTCTGCTAACATGG; Available on request from Steven Farber |

*Appendix 1 Continued on next page*

*Appendix 1 Continued*

| Reagent type (species) or resource | Designation | Source or reference | Identifiers | Additional information |
|---|---|---|---|---|
| Recombinant DNA reagent | Pk-GoldyTal(plin2 Pair 1 RightTAL 2)(plasmid) | This paper | | TALEN target: AAATAACCAGGTTTGCC; Available on request from Steven Farber |
| Recombinant DNA reagent | Pk-GoldyTal(plin2 Pair 2 LeftTAL 1)(plasmid) | This paper | | TALEN target: TTTCTGCTAACATGGGT; Available on request from Steven Farber |
| Recombinant DNA reagent | Pk-GoldyTal(plin2 Pair 2 RightTAL 2)(plasmid) | This paper | | TALEN target: AATAACCAGGTTTGCC; Available on request from Steven Farber |
| Recombinant DNA reagent | Pk-GoldyTal(plin3 Pair 1 LeftTAL 1) (plasmid) | This paper | | TALEN target: TTGCGCCTCAGATAAC; Available on request from Steven Farber |
| Recombinant DNA reagent | Pk-GoldyTal(plin3 Pair 1 RightTAL 2) (plasmid) | This paper | | TALEN target: AATTGCCACACAACCT; Available on request from Steven Farber |
| Recombinant DNA reagent | Pk-GoldyTal(plin3 Pair 2 LeftTAL 1) (plasmid) | This paper | | TALEN target:CAGATAACAGAGAAA; Available on request from Steven Farber |
| Recombinant DNA reagent | Pk-GoldyTal(plin3 Pair 2 RightTAL 2) (plasmid) | This paper | | TALEN target:CACACAACCTAAATA; Available on request from Steven Farber |
| Recombinant DNA reagent | pKT3Ts-goldyTALEN vector | *Welker et al., 2016* | Addgene plasmid #80330; RRID:Addgene_80330 | |
| Recombinant DNA reagent | FusX TALEN assembly system | *Ma et al., 2013*; *Ma et al., 2016* | Addgene kit #1000000063 | |
| Recombinant DNA reagent | pDestTol2Pa2(plasmid) | *Kwan et al., 2007* | Plasmid #394 | http://tol2kit.genetics.utah.edu |
| Recombinant DNA reagent | P4-P1R_ zPLIN2_5'UTR(plasmid) | This paper | | 5' entry plasmid for left homology arm of pDestTol2Pa2(EGFP-plin2) donor plasmid V2; Available on request from Steven Farber |
| Recombinant DNA reagent | pME-EGFP no stop 12aa linker(plasmid) | This paper | | EGFP middle entry plasmid for pDestTol2Pa2(EGFP-plin2) donor plasmid V2Includes strong zebrafish kozak sequence; Available on request from Steven Farber |
| Recombinant DNA reagent | P3R-P3-zPLIN2_1–900 | This paper | | 3' entry plasmid for right homology arm of pDestTol2Pa2(EGFP-plin2) donor plasmid V2; Available on request from Steven Farber |
| Recombinant DNA reagent | pDestTol2Pa2(EGFP-plin2) donor plasmid V2 | This paper | | Donor plasmid for creation of Fus(EGFP-plin2) fish |

*Appendix 1 Continued on next page*

*Appendix 1 Continued*

| Reagent type (species) or resource | Designation | Source or reference | Identifiers | Additional information |
|---|---|---|---|---|
| Recombinant DNA reagent | P4-P1R_zPLIN3_last679 | This paper | | 5' entry plasmid for left homology arm of pDestTol2Pa2(PLIN3-RFP); Available on request from Steven Farber |
| Recombinant DNA reagent | tagRFP-t coding sequence | *Horstick et al., 2015* | | Modified for zebrafish from Addgene #61390 |
| Recombinant DNA reagent | pME-tagRFP-t (plasmid) | This paper | | Middle entry plasmid for pDestTol2Pa2(PLIN3-RFP); Available on request from Steven Farber |
| Recombinant DNA reagent | P2R_P3_zPLIN3_3'UTR | This paper | | 3' entry plasmid for right homology arm of pDestTol2Pa2(PLIN3-RFP); Available on request from Steven Farber |
| Recombinant DNA reagent | pDestTol2Pa2(plin3-RFP) donor plasmid | This paper | | Donor plasmid for creation of Fus(plin3-RFP) fish; Available on request from Steven Farber |
| Recombinant DNA reagent | pDestTol2Pa2(fabp2: EGFP-PLIN2)(plasmid) | This paper | | Zebrafish fabp2 promoter, human PLIN2 coding sequence; Available on request from Steven Farber |
| Recombinant DNA reagent | pDestTol2Pa2(fabp10a: EGFP-PLIN2)(plasmid) | This paper | | Zebrafish fabp10a promoter, human PLIN2 coding sequence; Available on request from Steven Farber |
| Recombinant DNA reagent | pDestTol2Pa2(hsp70l: EGFP-PLIN2)(plasmid) | This paper | | Zebrafish hsp70l promoter, human PLIN2 coding sequence; Available on request from Steven Farber |
| Recombinant DNA reagent | pDestTol2Pa2(fabp2: PLIN3-EGFP)(plasmid) | This paper | | Zebrafish fabp2 promoter, human PLIN3 coding sequence; Available on request from Steven Farber |
| Recombinant DNA reagent | pDestTol2Pa2(hsp70l: PLIN3-EGFP)(plasmid) | This paper | | Zebrafish hsp70l promoter, human PLIN3 coding sequence; Available on request from Steven Farber |
| Recombinant DNA reagent | pGFP-hADRP(plasmid) (plasmid) | *Targett-Adams et al., 2003* | | Human PLIN2 coding sequence (Alternative name ADRP) in pEGFP-C1 vector |
| Recombinant DNA reagent | pCR8(pGFP-hADRP) (plasmid) | This paper | | Human PLIN2 coding sequence (Alternative name ADRP); Available on request from Steven Farber |
| Recombinant DNA reagent | pDNR-Dual(h Perilipin 3) (plasmid)(TIP47) | FLEXgene Repository (Harvard Medical School) | Clone ID:HsCD00004695 | Human PLIN3 coding sequence Alternative name: TIP47 |
| Recombinant DNA reagent | pCR8(Perilipin3)(TIP47) (plasmid) | This paper | | Middle entry plasmid, Human PLIN3 coding sequence Alternative name: TIP47; Available on request from Steven Farber |

*Appendix 1 Continued on next page*

*Appendix 1 Continued*

| Reagent type (species) or resource | Designation | Source or reference | Identifiers | Additional information |
|---|---|---|---|---|
| Recombinant DNA reagent | Zebrafish fabp2 promoterp5E fabp2 (–2.3 kb) fabp2 (plasmid) | *Park et al., 2019* | | 5' entry plasmid, zebrafish fabp2 (ifabp) promoter sequence |
| Recombinant DNA reagent | p5E fabp10a, (–2.8 kb) | *Her et al., 2003* | | zebrafish fabp10a (lfabp) promoter sequence |
| Recombinant DNA reagent | p5E-hsp70lheat shock cognate 70 kDa protein, like | *Kwan et al., 2007* | Plasmid #222 | http://tol2kit.genetics.utah.edu |
| Recombinant DNA reagent | p3E-EGFPpA | *Kwan et al., 2007* | Plasmid #366 | http://tol2kit.genetics.utah.edu |
| Recombinant DNA reagent | pME-EGFP no stop | *Kwan et al., 2007* | Plasmid #455 | http://tol2kit.genetics.utah.edu |
| Recombinant DNA reagent | p3E-polyA | *Kwan et al., 2007* | Plasmid #302 | http://tol2kit.genetics.utah.edu |
| Sequence-based reagent | PCR primers | This paper | | Please refer to *Supplementary file 1* |
| Commercial assay or kit | T3 Message Machine kit | Thermo Fisher Scientific | Catalog #AM1348 | |
| Commercial assay or kit | DIG RNA labelling kit | Roche | Catalog#11277073910 | |
| Other | T7 polymerase | Roche | Catalog #10881767001 | |
| Other | SP6 polymerase | Roche | Catalog #10810274001 | |
| Commercial assay or kit | TA Cloning Kit, Dual Promoter, with pCRII-TOPO vector | Thermo Fisher Scientific | Catalog # K207020 | |
| Commercial assay or kit | REDExtract-N-Amp Tissue PCR kit | Sigma-Aldrich | Catalog #XNAT | |
| Commercial assay or kit | RNA Clean and Concentrator Kit | Zymo Research | Catalog #R1013 | |
| Commercial assay or kit | iScript cDNA Synthesis Kit | Bio-Rad Laboratories, Inc. | Catalog #1708891 | |
| Commercial assay or kit | SsoAdvanced Universal SYBR Green Supermix | Bio-Rad Laboratories, Inc. | Catalog #1725271 | |
| Commercial assay or kit | MultiSite gateway assembly system | Invitrogen | Catalog #12537–023 | |
| Commercial assay or kit | pCR8/GW/TOPO TA cloning kit | Thermo FisherScientific | Catalog # K250020 | |
| Other | BODIPY (558/568)-C12 | Thermo Fisher Scientific | Catalog #D3835 | 4 µg/ml |
| Other | BODIPY FL-C12 | Thermo Fisher Scientific | Catalog #D3822 | 4 µg/ml |
| Other | Cholesterol | Sigma-Aldrich | Catalog#C8667 | 4% w/w |
| Other | HSC LipidTOX Green | Thermo Fisher Scientific | Catalog #H34475 | 1:5,000 |
| Other | HSC LipidTOX Red | Thermo Fisher Scientific | Catalog #H34476 | 1:5,000 |

*Appendix 1 Continued*

| Reagent type (species) or resource | Designation | Source or reference | Identifiers | Additional information |
|---|---|---|---|---|
| Other | RNA*later* | Thermo Fisher Scientific | Catalog # AM7020 | |
| Other | FokI restriction enzyme | New England Biolabs | Catalog # R0109 | |
| Other | HphI restriction enzyme | New England Biolabs | Catalog # R0158 | |
| Other | Gemma Micro 75, 150 & 500 | Skretting | | Zebrafish diet |
| Software, algorithm | Mojo Hand design tool | *Neff et al., 2013* | | |
| Software, algorithm | Fiji | *Schindelin et al., 2012* | (Fiji, RRID:SCR_002285) | |
| Software, algorithm | ilastik | *Berg et al., 2019* | Ilastik, RRID:SCR_015246 | |
| Software, algorithm | GraphPad Prism 9 | GraphPad | GraphPad Prism, RRID:SCR_002798 | |

