## [Decision Letter]

**Acceptance summary:**

Your revised manuscript addresses most of the requests of the reviewers; your rationale for not performing detailed fasting studies is acceptable, in light of the heat shock issue and the ubiquitous transgenic paper you cited. The quantification of data Figures 1 and 4; and the inclusion of standard length makes your resource of broader utility to the field in that it allows comparison with other studies. Likewise, the additional details on the knock-in strategy provide greater transparency and validation of your tool. You rightly note some errors in selecting citations for revision-thank you for this description of your choices. Your suite of knock-in and transgenic tools will serve as a resource of broad utility.

**Decision letter after peer review:**

Thank you for submitting your article "Imaging cytoplasmic lipid droplets in vivo with fluorescent perilipin 2 and perilipin 3 knockin zebrafish" for consideration by *eLife*. Your article has been reviewed by 3 peer reviewers, including Amnon Schlegel as the Reviewing Editor and Reviewer #1, and the evaluation has been overseen by Didier Stainier at Senior Editor.

The reviewers have discussed their reviews with one another, and the Reviewing Editor has drafted this single summary and list of essential revisions to help you prepare a revised submission.

Summary:

The authors sought to develop fluorescent protein reports for lipid droplets for use in live imaging of an intact vertebrate organism. These subcellular structures are under dynamic regulation by assorted intrinsic and extrinsic factors; their study would be enhanced by having a convenient, in vivo detection system. The authors present knock-in lines of two lipid droplet coat proteins (and several transgenic lines with restricted expression) toward this end. These tools could be valuable to the cell biological and metabolic research communities. The authors find that plin2 transcript is induced in liver and intestine of 6 dpf zebrafish larvae following a single feeding, while plin3 transcript is expressed in the fasted and fed states in the liver. They use TALENS to knock-in EGFP and TagRFP1 into the plin2 and plin3 loci, with the encoded gene products being the fusion proteins Plin2-EGFP and Plin3-TagRFP1. The Plin2-EGFP protein shows greater induction of fluorescence following a meal. The overall aim of these initial expression characterizations and development of lipid droplet reporter knock-ins is to be able to monitor the life cycle of these organelles in a living whole organism.

Higher resolution photomicrographs of lipid droplets with these knock-in lines concurrently stained with the fluorescent lipid dyes BODIPY C12 and BODIPY FL-C12 are presented with a time series following feeding in intestine; additional cell types beyond enterocytes (i.e., hepatocytes, adipocytes, and cells surrounding lateral line structures) are presented.

The authors are in a position to provide a technical advance to the field of lipid droplet biology and, potentially, assorted diseases marked by lipid derangements. With the tractable revisions set out below, their tools set the stage for chemical and genetic screens for factors and compounds that modulate the normal life cycle of these dynamic organelles.

Essential revisions:

1) Please present data with greater quantitative rigor throughout the manuscript.

a. Figure 1E and 1F, for instance, require quantification of the effects of induction. More generally, the number of animals studied (5 to 10) per experimental time point was felt to be low given the large number of embryos obtained in a cross, and the potential to reveal whether lipid droplet (coat protein signal) is gaussian in distribution or has a threshold; previous work by the Rawls and Sadler labs have pointed to standard length as the driver (not age) of detecting lipid droplets with fluorescent (and other) dyes.

b. Relatedly, statistical rigor in these analyses is requested; clear statements throughout of what tests are used is required.

2) Please improve the description and validation of the knock-in and transgenic lines.

a. Why was the 5'-end used to insert one fluorescent reporter, but the 3'-end was used for the other? Please describe (based on cell biological and *C. elegans* literature) whether this choice matters. Does this merely reflect the cDNAs obtained from others? Was there a deliberate strategy?

b. Were insertions or deletions introduced beyond the sites of knock-in? Are there off-targets?

c. Since GFP and TagRFP-T are not fast folding proteins and are very stable, a kinetic analysis directly comparing lipid dye signals to fluorescent proteins in both growing and shrinking lipid droplets is required. For instance, please provide fluorescent dye labeling of cell types beyond enterocytes in Figure 2. Any cases where this is not feasible can be informative. For example, if the live liver is not well-labeled, then the transgenic reporters will clearly provide an advantage for imaging.

d. Relatedly, (i) please move Figure Supplement 1 relating to the assorted transgenic lines to the main figures and include co-labeling with a fluorescent lipid dye, and (ii) please provide whole-mount, low magnification views of adults subjected to a fast along the lines of the studies conducted by Minchin and Rawls. This experiment will address similar issues to that raised in points 2c and 2e.

e. For the transgenic lines, please state why human orthologs were selected? For instance, if you can detect these with commercially available antibodies, that could be a major advantage to their use. While PLIN2 and 3 are generally considered permissive to the lipolytic cascade (ATGL, HSL, and assorted co-factors) without hormonal input, it is important to assess whether the over-expressed proteins impact (i.e., prevent lipolysis) lipid droplet biology. A fasting series of up to 7 days in larvae is feasible and might reveal neomorphic aspects to these transgenics.

3) Please provide broader context regarding the value of these new tools in a dedicated Discussion section.

The summary paragraph (lines 331-340) should be expanded to describe the potential uses of these new reagents for studying obesity and lipodystrophy, along with fatty liver disease (whether alcoholic, non-alcoholic, or in-born error of metabolism-related), diabetes mellitus (types 1 and 2), specific cardiovascular diseases and atherosclerosis (e.g., are you talking about potential for imaging macrophages in plaques)? More narrowly, please describe how these disease processes would be explored in zebrafish: are there established dietary models of fatty liver disease, type1 or type 2 diabetes, or atherosclerosis in which these tools could be used? Are there genetic or chemical screens you foresee would benefit from these lines? Could they be crossed to established mutants for live imaging purposes?

Paralleling this disease modeling discussion, please discuss the limitations of using a perilipins on lipid droplet biogenesis investigations: early lipid droplet biogenesis proceeds without perilipins, discussion of what other tagged proteins might be required for visualizing those early steps in a whole vertebrate also merits presentation.

4. Improve citations to primary literature or key review articles of direct relevance to the Tools and Resources format. Original citations can be added to address point 3.

Please remove the following citations, unless you feel a particular review article raises a unique hypothetical or synthetic point:

1. Bickel et al.,

2. Chien et al., (although this method's use in zebrafish has recently been reported and the authors should cite: https://www.nature.com/articles/s41467-020-19827-1 );

3. Kimmel and Sztalryd.

4. Mather et al.,

5. Miura et al.,

6. Olzmann and Carvalho (if a review article regarding lipid droplets is felt to be needed, this or the next citation is the most recent one and could remain).

7. Roberts and Olzmann (see previous point; the Coleman 2020 review might suffice, in general, alternatively).

8. Sztalryd and Brasaemle

9. Thomas et al., 2015

10. Zeituni and Farber

---

## [Author Response]

Essential revisions:1) Please present data with greater quantitative rigor throughout the manuscript.a. Figure 1E and 1F, for instance, require quantification of the effects of induction. More generally, the number of animals studied (5 to 10) per experimental time point was felt to be low given the large number of embryos obtained in a cross, and the potential to reveal whether lipid droplet (coat protein signal) is gaussian in distribution or has a threshold; previous work by the Rawls and Sadler labs have pointed to standard length as the driver (not age) of detecting lipid droplets with fluorescent (and other) dyes.b. Relatedly, statistical rigor in these analyses is requested; clear statements throughout of what tests are used is required.

Based on reviewer requests, we have provided more quantitation throughout the manuscript. In particular, we quantified the effects of induction associated with Figure 1E at both the whole mount and sub-cellular level and in doing so increased the numbers of animals studied. These new analyses can be found in Figure 2 (whole mount, 14-23 fish at each of 20 time-points) and Figure 4 (sub-cellular, 3-18 fish at each of 21 time-points) in the revised manuscript. With regard to the number of animals studied, during a confocal time-course, the number of animals used in the analysis is dependent on how many fish we can mount and image at a given time-point. In the first hours after feeding, our time-points are closer together, so fewer fish are imaged. Furthermore, we can sort for Plin3-RFP before feeding, but it is not possible to do so for the EGFP-Plin2 line. Typically, we have imaged at least 2x as many fish as noted in the figure legend and it is only after collecting and genotyping the imaged larvae that we know how many EGFP-Plin2 fish were present. At later time-points when the EGFP signal is visible, it is easier to ensure that we have higher numbers of fish. While we could repeat these time-courses additional times to increase the numbers of larvae, these are very time-intensive experiments and we feel that the data we have collected in Figure 4 is more than sufficient to show the clear transition from Plin3 to Plin2 on the lipid droplets after a meal. As requested, we have also performed and reported statistical analyses where appropriate.

With regard to the use of standard length and not age, for any imaging performed in the liver, adipose tissue and lateral line, we have now provided standard lengths of the fish in addition to the day post fertilization at which the imaging was performed. For images provided in the initial submission for which no standard length measurements were available, as a proxy, we have provided standard length data for fish that were treated similarly with regard to food and rearing.

For Example, see Figure 5 legend:

“Standard length of the imaged fish is noted on the upper left corner of each set of images, see source data for additional standard length data.”

2) Please improve the description and validation of the knock-in and transgenic lines.a. Why was the 5'-end used to insert one fluorescent reporter, but the 3'-end was used for the other? Please describe (based on cell biological and *C. elegans* literature) whether this choice matters. Does this merely reflect the cDNAs obtained from others? Was there a deliberate strategy?

We have addressed in the text why we chose to insert EGFP at the 5’ end and RFP at the 3’ end. The decision for the 5’ EGFP on Plin2 was based on an early report that only N-terminal fusions resulted in functional proteins (Targett-Adams et al., 2003). Since we began the synthesis of these lines, C-terminal fusions have now also been shown to be functional. The tags were placed on the C-terminal end of Plin3 because our human transgenic lines with C-terminal fusions were known to be functional before we made the knock-in line.

Results section now reads:

“We designed our engineering strategy for Fus(EGFP-plin2) based on early data suggesting that it was necessary to tag PLIN2 on the N-terminus (Targett-Adams et al., 2003), although subsequent work has now shown that C-terminal tags are also functional (Kaushik and Cuervo 2015, Lumaquin et al., 2021). Additionally, we had evidence from our generation of Tol2-based transgenic reporter lines that over-expression of both human EGFP-PLIN2 and PLIN3-EGFP resulted in labeling of lipid droplets (refer to Figure 4 —figure supplement 1 and Figure 8).”

b. Were insertions or deletions introduced beyond the sites of knock-in? Are there off-targets?

We have now included additional details about the numbers of fish screened to find our knock-in lines. We noted the incorrect insertions discovered and the one intronic deletion that we found in the plin3 line.

Results section now reads:

“For Fus(EGFP-plin2), a total of 40 adults were screened, and 3 fish produced EGFP+ progeny. However, two of these clutches had abnormal tissue expression patterns and only displayed cytoplasmic EGFP fluorescence. One of these had correct integration across the left homology arm, but the right homology arm integration could not be verified; the second had no confirmed integration at the locus. Furthermore, while our founder fish produced low numbers of progeny with correct expression patterns and proper integration (8 embryos out of 853 screened), she also produced progeny with incorrect integration (22/853), suggesting mosaicism in the germ cells. To identify a Fus(plin3-RFP) founder, 26 adults were screened before a fish produced progeny expressing RFP. These embryos had correct tissue expression of RFP in the intestine and correct integration of the transgene (53 RFP+ embryos out of 613 screened). We did note a 29 bp region missing in intron 7-8, however, this does not alter the coding region, and it is unclear whether this deletion arose during integration or whether it was a naturally occurring variant in the injected embryo.”

c. Since GFP and TagRFP-T are not fast folding proteins and are very stable, a kinetic analysis directly comparing lipid dye signals to fluorescent proteins in both growing and shrinking lipid droplets is required. For instance, please provide fluorescent dye labeling of cell types beyond enterocytes in Figure 2. Any cases where this is not feasible can be informative. For example, if the live liver is not well-labeled, then the transgenic reporters will clearly provide an advantage for imaging.

We have now provided extensive time-course imaging and kinetic analysis of Plin3-RFP and EGFP-Plin2 in the intestine in larvae fed with fluorescent BODIPY fatty acids (refer to new panels in Figure 4 and associated supplemental figures and videos). We imaged both the induction (0 – 8.5 h) and the degradation phase (16 – 30 h) of the lipid droplets. We find that at all times, the BODIPY-labeled lipid droplets in the intestine are labeled with either Plin3 or Plin2.

Unfortunately, this analysis does not allow us to directly determine whether the EGFP or TagRFP-t tags affect the folding or stability of the Plin proteins compared to untagged versions (and thus possibly influence lipid droplet growth and lifetime). Since we do not have antibodies that recognize the zebrafish plin proteins, this direct comparison is not possible at this time.

The text of the Discussion now reads:

“Therefore, we have little reason to believe the fluorescent tags used in this study are interfering with lipid droplet dynamics, however we cannot test these hypotheses directly due to the current lack of antibodies that recognize the zebrafish perilipin 2 and 3 orthologs. Despite these caveats, these reporters indicate a clear progression from Plin3 to Plin2 on intestinal lipid droplets following a high-fat meal in zebrafish.”

As requested, we have now provided additional images of fluorescent LipidTOX labeling of the lipid droplets in fish expressing the fluorescent Plin proteins (see Figures 5, 6, and 7).

d. Relatedly, (i) please move Figure Supplement 1 relating to the assorted transgenic lines to the main figures and include co-labeling with a fluorescent lipid dye, and (ii) please provide whole-mount, low magnification views of adults subjected to a fast along the lines of the studies conducted by Minchin and Rawls. This experiment will address similar issues to that raised in points 2c and 2e.

i) We have moved the figure supplement to the main text (Figure 8) and we have now included co-labeling with a fluorescent lipid dye for each transgenic line.

ii) We did not generate our fusion lines in a pigment mutant background, so whole-mount imaging of the perilipins in the adult pigment mutant fish would require many months of breeding. However, in Figure 6, we have now provided whole-mount images of EGFP-Plin2 and Plin3-RFP in the adipose tissue in juvenile larvae (21dpf, standard length is noted in the figure panel, lipid droplets are also labeled with LipidTOX dyes).

It was unclear whether the reviewers were requesting the fasting experiment in the knock-in lines or in the over-expression transgenic lines. We did not perform a fasting experiment with our fusion lines because in the recent Lumaquin et al., eLife 2021 study, stable over-expression Plin2-tdtomato did not show differences in adipose area, standard length or adipose area/standard length following a 7 day fast. Given that our knock-in lines are expressing the fusion proteins at endogenous levels, it is therefore unlikely that the fluorescent tags would cause alterations in lipolysis. The Tol2 transgenic lines we have generated over-express human PLINs either in the intestine, liver or ubiquitously following heat-shock. Therefore, to have overexpression of PLIN2 or PLIN3 in the adipocytes, a daily heat-shock would likely be required and would add additional stress to the fish being starved. While it is feasible that over-expression of PLIN2 in the liver may alter the effects of a fast on adipocytes or other tissues in some way, it is unlikely that over-expression in the intestine would have any effect if the fish is not consuming any dietary lipids and have no lipid droplets in the intestine. While these are interesting questions and could be pursued in the future, we feel that they are not essential to the description/validation of the new tools that we are highlighting in the current manuscript.

e. For the transgenic lines, please state why human orthologs were selected? For instance, if you can detect these with commercially available antibodies, that could be a major advantage to their use. While PLIN2 and 3 are generally considered permissive to the lipolytic cascade (ATGL, HSL, and assorted co-factors) without hormonal input, it is important to assess whether the over-expressed proteins impact (i.e., prevent lipolysis) lipid droplet biology. A fasting series of up to 7 days in larvae is feasible and might reveal neomorphic aspects to these transgenics.

We have included information in the text about why we chose to over-express the human orthologs and why they may provide an advantage over the zebrafish orthologs.

The Results section now reads:

“The decision to over-express the human orthologs was mostly out of convenience of already having the clones present in our laboratory and evidence from cell culture studies that fusions of human PLIN proteins with fluorescent proteins were functional (Targett-Adams et al., 2003). However, we also appreciated that human proteins are more amenable to detection with commercially available antibodies in downstream applications, and might provide an opportunity to maintain tissue-specific or temporal control of perilipin expression when targeting the endogenous loci for gene editing.”

To begin to address whether the over-expressed proteins impact lipid droplet dynamics, we performed the kinetic BODIPY C12 intestinal time-course imaging in the Tg(fabp2:EGFP-PLIN2) transgenic line alongside the fusions lines (Figure 4). This allowed us to ask whether the lipid droplets are amenable to PLIN2 binding if the protein is present earlier in the enterocyte and to ask whether the lifetime of the lipid droplets was increased due to the presence of the over-expressed human EGFP-PLIN2 when compared with endogenously-expressed zebrafish EGFP-Plin2. These experiments indicate that when expressed under the control of the fabp2 promoter, the human PLIN2 protein is present earlier in the enterocyte following onset of a high-fat meal and localizes primarily to the lipid droplets. Thus, the nascent lipid droplets are amenable to PLIN2 binding. However, we did not find a significant difference in the % area occupied by lipid droplets at 30 h. At this time, fish of each genotype had a range of lipid droplets remaining, some had none and whereas others had droplets occupying upwards of 15-20% of the intestine area, the mean for all genotypes was 6.2%. Furthermore, at 6 months of age, we did not find differences between the Tg(fabp2:EGFP-PLIN2)/+ fish and their WT siblings in either standard length or mass, suggesting that the subtle differences noted in the intestine did not have lasting impacts on their physiology.

The following paragraph has been added to the results:

“Based on this analysis, we wondered whether the timing of the transition to Plin2 simply reflects the time needed to both transcribe and translate Plin2 prior to its association with the droplets, and whether Plin2 could bind to nascent lipid droplets sooner if the protein was already present. To investigate this question, we repeated the intestinal time-course imaging in transgenic fish stably over-expressing human PLIN2 under the control of the zebrafish fatty acid binding protein 2 promoter, Tg(fabp2:EGFP-PLIN2). This promoter drives expression in the yolk syncytial layer and in the intestine (Her et al., 2003)(Figure 4 —figure supplement 1). In unfed fish, EGFP-PLIN2 signal cannot be detected above the autofluorescence visible in the intestine at 6 dpf (Figure 4A Unfed, Figure 4 —figure supplement 1), likely because any protein produced is rapidly degraded in the absence of lipid droplets (Xu et al., 2005). However, we hypothesized that the mRNA would be available immediately for translation and PLIN2 protein would be produced more quickly than in the Fus(EGFP-plin2) line. Indeed, we visually detected EGFP-PLIN2 on lipid droplets at 45 min, our earliest time-point after the start of the meal (Figure 4A and E) and the difference between fluorescence on lipid droplets vs. in the cytoplasm reached statistical significance by 3.5 h (p = 0.0134; Two-way ANOVA with Šídák multiple comparisons test), suggesting that the droplets are not prohibitive to plin2 binding at the early times. Although we also hypothesized that over-expression of EGFP-PLIN2 might delay the degradation of lipid droplets in the enterocytes, the fraction of intestine area covered by remaining lipid droplets at 30 h was not different between the three transgenic lines or wild-type fish (p = 0.2922, Kruskal-Wallis test, n = 9–23 fish, Figure 4 —figure supplement 2). Furthermore, over-expression of EGFP-PLIN2 in the intestine does not affect the standard length or mass of the fish at 6 months of age (Figure 4 —figure supplement 1).”

As discussed above, the tissue in which our human transgenic lines would be most appropriate to be used in a fasting series would be the liver. While we did perform short periods of fasting (48 – 72 h) in the Tg(fabp10a:EGFP-PLIN2) line, we could not draw sufficient conclusions from these experiments to warrant addition in this manuscript. Although we expected to find that these fish had many more lipid droplets in the liver than wild-type siblings or the fusion lines subjected to the same period of fasting, the variability between fish in lipid droplet number and size, irrespective of the genotype of the fish, made it very difficult to interpret the effects of fasting and/or over-expression. We feel strongly that many further experiments are necessary to even begin to understand lipid metabolism and perilipins in the developing liver.

3) Please provide broader context regarding the value of these new tools in a dedicated Discussion section.The summary paragraph (lines 331-340) should be expanded to describe the potential uses of these new reagents for studying obesity and lipodystrophy, along with fatty liver disease (whether alcoholic, non-alcoholic, or in-born error of metabolism-related), diabetes mellitus (types 1 and 2), specific cardiovascular diseases and atherosclerosis (e.g., are you talking about potential for imaging macrophages in plaques)? More narrowly, please describe how these disease processes would be explored in zebrafish: are there established dietary models of fatty liver disease, type1 or type 2 diabetes, or atherosclerosis in which these tools could be used? Are there genetic or chemical screens you foresee would benefit from these lines? Could they be crossed to established mutants for live imaging purposes?Paralleling this disease modeling discussion, please discuss the limitations of using a perilipins on lipid droplet biogenesis investigations: early lipid droplet biogenesis proceeds without perilipins, discussion of what other tagged proteins might be required for visualizing those early steps in a whole vertebrate also merits presentation.

As suggested by the Reviewers, we have now extensively expanded the Discussion section to provide broader context as to the value of these new tools and to further discuss our findings regarding the ordered recruitment of plin3 and plin2 in the intestine following a high-fat meal. We specifically discuss the need to study lipid droplet biology in the context of whole animal studies.

For example, here is a section from the penultimate paragraph of the Discussion:

“Using these zebrafish Plin reporter lines in the context of diets (Stoletov et al., 2009, Turola et al., 2015, Sapp et al., 2014) and established zebrafish mutations and disease models (Holtta-Vuori et al., 2013, Maddison et al., 2015, Liu et al., 2015, Liu et al., 2018, O'Hare et al., 2014) may provide mechanistic insights that connect perilipin cell biology to metabolic and cardiovascular diseases.”

4. Improve citations to primary literature or key review articles of direct relevance to the Tools and Resources format. Original citations can be added to address point 3.Please remove the following citations, unless you feel a particular review article raises a unique hypothetical or synthetic point:1. Bickel et al.,2. Chien et al., (although this method's use in zebrafish has recently been reported and the authors should cite: https://www.nature.com/articles/s41467-020-19827-1 );3. Kimmel and Sztalryd.4. Mather et al.,5. Miura et al.,6. Olzmann and Carvalho (if a review article regarding lipid droplets is felt to be needed, this or the next citation is the most recent one and could remain).7. Roberts and Olzmann (see previous point; the Coleman 2020 review might suffice, in general, alternatively).8. Sztalryd and Brasaemle9. Thomas et al., 201510. Zeituni and Farber

We have augmented many of the reviews with primary literature, however, there were a number of primary references in the above list, so we have addressed each reference request individually below:

1. The Bickel et al., 2009 review provides important comparisons amongst the plin family proteins that are not readily appreciated when reading all of the original primary research articles and additionally provides relevant historical context. However, we have removed this reference to satisfy the reviewers’ request.

2. The Chien et al., 2012 reference is a primary article using CARS microscopy to elucidate lipid droplets within the *Drosophila* model system. While the reference noted by the reviewers describes the use of confocal Raman spectroscopic imaging in zebrafish and briefly describes lipid droplets, we feel that this suggested reference does not adequately make the point that CARS and SRS microscopy can be used to image lipid droplets and should not replace our original reference. Furthermore, the sentence in which our reference occurs is not strictly describing zebrafish model systems.

3. Kimmel and Sztalryd 2016 – We have added a number of primary references to support our statements, however, given the complexity of the PLIN protein family (and all of the various names each of the family members have had over the years) we feel that this review article is still an important addition to this manuscript for readers who may not be as familiar with the lipid droplet/perilipin field (especially zebrafish researchers who might want to utilize our knock-in lines).

4. We have replaced the Mather et al., 2019 symposium review article reference with the Masedunskas et al., 2017 article, which is the source of the data provided in the Mather et al., 2019 symposium paper.

5. Miura et al., 2002. We do not understand why the reviewers have requested this reference removed. This is a primary article which beautifully shows fluorescently-tagged perilipin proteins from different species associating with lipid droplets in cultured cells. It is also the first study to show that proteins identified as perilipins in other species based on sequence identity, associate with lipid droplets.

6 and 7. Olzmann and Carvalho, 2019 and Roberts and Olzmann, 2020. These reviews are some of the most recently written on lipid droplets and provide a solid background for readers less familiar with the lipid droplet field. Together, they provide a thorough review of lipid droplet functions and dynamics, including budding and growth, organelle contacts, and roles in lipotoxicity and ER stress, as well as the role of the lipid droplet proteome in these dynamics and functions. If we were to try to replace these reviews, it would require hundreds of primary references. Given that the role of these reviews was to introduce readers to the fact that lipid droplets are not just bags of fat, but that they are important organelles with many roles in physiology, we feel that they accomplish this purpose more completely than a handful of randomly chosen primary references. To reduce the reviews cited in this section of the introduction, we have removed the Coleman 2020 reference because this is not as comprehensive a review as the Olzmann and Carvalho, 2019 and Roberts and Olzmann, 2020 references.

8. Sztalryd and Brasaemle, 2017. We have added a number of additional primary references to this section of the introduction, but we feel strongly that this review concisely summarizes our understanding of the regulation of lipolysis by the perilipins and think that it should remain in the paper.

9. Thomas et al., 2015. We have replaced this reference with more appropriate primary references.

10. Zeituni and Farber 2016 is a methods paper that provides step-by-step protocols for a reader interested in repeating the feeding and imaging assays described in this manuscript. Given eLife’s commitment to enhancing data reproducibility, we think that this reference should remain in our Methods and Materials section.